# SSC: The novel self-stack ensemble model for thyroid disease prediction

**Shengjun Ji** [iD] *

School of information, Xi'an University of Finance and Economics, Xi'an, China

* xcshare@163.com

**Citation:** Ji S (2024) SSC: The novel self-stack ensemble model for thyroid disease prediction. PLoS ONE 19(1): e0295501. https://doi.org/10.1371/journal.pone.0295501

**Data Availability Statement:** https://archive.ics.uci.edu/ml/datasets/thyroid+disease.

**Funding:** First-class 431 Course Construction Based on Online and Offline Xi´an University of Finance and Economics (Object-Oriented Technology and Programming) [2020].

## Abstract

Thyroid disease presents a significant health risk, lowering the quality of life and increasing treatment costs. The diagnosis of thyroid disease can be challenging, especially for inexperienced practitioners. Machine learning has been established as one of the methods for disease diagnosis based on previous studies. This research introduces a novel and more effective technique for predicting thyroid disease by utilizing machine learning methodologies, surpassing the performance of previous studies in this field. This study utilizes the UCI thyroid disease dataset, which consists of 9172 samples and 30 features, and exhibits a highly imbalanced target class distribution. However, machine learning algorithms trained on imbalanced thyroid disease data face challenges in reliably detecting minority data and disease. To address this issue, re-sampling is employed, which modifies the ratio between target classes to balance the data. In this study, the down-sampling approach is utilized to achieve a balanced distribution of target classes. A novel RF-based self-stacking classifier is presented in this research for efficient thyroid disease detection. The proposed approach demonstrates the ability to diagnose primary hypothyroidism, increased binding protein, compensated hypothyroidism, and concurrent non-thyroidal illness with an accuracy of 99.5%. The recommended model exhibits state-of-the-art performance, achieving 100% macro precision, 100% macro recall, and 100% macro F1-score. A thorough comparative assessment is conducted to demonstrate the viability of the proposed approach, including several machine learning classifiers, deep neural networks, and ensemble voting classifiers. The results of K-fold cross-validation provide further support for the efficacy of the proposed self-stacking classifier.

## 1 Introduction

Thyroid disease diagnosis is a challenging and time-consuming procedure that requires substantial knowledge and experience [1]. Two common methods for diagnosis are doctors' examinations and a multitude of blood tests. Early detection of thyroid disease is highly desirable due to its wide-ranging symptoms, including difficulties in losing weight, obesity, constipation, muscle pain, hypersensitivity to colds, fatigue, and exhaustion. Moreover, thyroid disease affects a significant portion of the global population, with 12% of the US population experiencing the disease at some point in their lives according to the American Thyroid

**Competing interests:** The authors have declared that no competing interests exist.

Association (ATA) [2]. Therefore, an accurate and prompt disease identification system is essential to avoid major risks to the patient.

Machine learning has completely transformed thyroid disease detection and identification by making optimal use of time and decreasing misdiagnoses brought on by human mistakes. With the use of machine learning algorithms, efficient techniques for knowledge discovery and classification in thyroid disease datasets can be established [3]. For instance, Raisinghani et al. [4] proposed a machine learning-based prediction model for thyroid disease detection using Decision Trees (DT). Similarly, Tyagi et al. [5] estimated the probable risk of thyroid disease using conventional machine learning classifiers. Researchers in [6] evaluated the potential of machine learning classifiers such as K-Nearest Neighbours (KNN), Support Vector Machine (SVM), and Random Forest (RF) in the prediction of thyroid disease. Several other researchers collaborated to develop an effective machine learning-based approach for detecting thyroid disease [7–10]. However, the proposed systems incorporated machine learning algorithms that operated the detection of disease in an individual manner. An ensemble model, on the other end, has been established to be a promising strategy to increase the stability and accuracy of a classifier [11]. Numerous studies have been conducted utilizing various techniques to create efficient ensembles. Stacking is one of the most effective approaches for integrating classifiers and enhancing prediction accuracy [12, 13].

This research suggests a systematic approach for the reliable and efficient diagnosis of thyroid disease so that medical professionals can benefit from the advancements in computer science research. We focus on developing an efficient stacked ensemble algorithm that can provide highly accurate and reliable predictions for thyroid disease detection. Traditional stacking algorithms employ a collection of heterogeneous base learners and a single metalearner. Base learners are machine learning algorithms that produce results after being trained on dataset attributes. The meta-learner is a machine learning classifier that determines the best approach to integrate those output predictions to generate the final outcome. Since the performance of different machine learning algorithms varies, integrating the prediction of different algorithms optimizes classification performance. In contrast to heterogeneous base learners, this study uses a self-stacking ensemble in which the output predictions of four distinct RF variants as base learners are combined and delivered to another RF variant functioning as a metalearner. RF is useful for high-dimensional problems with highly correlated attributes and has a high prediction accuracy, especially in the context of medical diagnosis that frequently arises in bio-informatics [14–16].

Another problem associated with disease diagnosis is the imbalanced distribution of disease samples in the dataset, which could produce misleading and unreliable results. This is because machine learning models learn the decision boundary for the majority class more efficiently than for the minority class. Therefore, one of the most well-known approaches to classifying an imbalanced dataset is to modify the dataset composition [17]. To achieve this, we downsample the dataset to generate an equal number of samples for each target variable. This will produce ample data for each target variable needed for algorithm training.

Our study's uniqueness can be summed up as follows:

- The resampling technique was applied to address data discrepancies. Down-sampling was used in the majority of classes, resulting in a balanced distribution of samples across each target variable.

- The behavior of machine learning models in the field of thyroid disease data was investigated, which is a significant new area to consider in terms of imbalanced data.

- A randomized tree-based self-stacking ensemble model was developed for highly accurate detection of thyroid disease.

- The effectiveness of the proposed approach was demonstrated by comparing its performance with that of other machine learning, ensemble, and deep learning models.

- While previous studies mainly focused on binary classification for thyroid disease detection [18–21], our objective was to devise a multi-class classification system for detecting thyroid disease.

The remainder of the article is structured as follows: After a brief introduction to the underlying research, Section 2 discusses the background of thyroid disease and its previously presented detection methods. The description of the methodology used for this study's experimental approach is provided in Section 3. In Section 4, results from extensive experiments are reported and discussed. A comparative analysis of the results is presented in Section 5. Finally, Section 6 draws the research to a conclusion.

## 2 Background

The primary cause of thyroid disease is an imbalance in the hormones generated by the thyroid gland. Euthyroidism, hyperthyroidism, and hypothyroidism, which pertain to normal, excessive, or deficient thresholds of thyroid hormones, are the rationale for diagnosing thyroid disease. Euthyroidism describes the thyroid gland's normal thyroid hormone production and cellular thyroid hormone levels [22]. The clinical manifestation of hyperthyroidism is increased circulatory and intracellular thyroid hormones [23]. Hypothyroidism is caused mostly by a deficiency of thyroid hormone production and inadequate alternative therapy [24]. Albeit, thyroid disease detection appears to be a basic recurring activity for medical experts, there is an unmet need to facilitate the reader in diagnosing thyroid disease with greater accuracy and consistency [25]. Moreover, the treatment of disease is a continuing problem for medical professionals, and accurate and timely diagnosis is crucial.

Recently, machine learning and deep learning demonstrated notable advances in the diagnosis of thyroid disease. Sidiq et al. [26], for example, used different machine learning models, including K-Nearest Neighbors (KNN), Decision Tree (DT), Naive Bayes (NB), and Support Vector Machine (SVM), to diagnose thyroid disease from a clinical dataset collected from a medical facility in Kashmir. The dataset included 553 data samples from healthy individuals, 218 samples from hypothyroid patients, and 36 samples from hyperthyroid patients. The authors conducted classification and reported the highest accuracy score of 98.89% yielded by DT. Another study [27], evaluated the potential of three machine learning models including KNN, DT, and Logistic Regression (LR) for the diagnosis of thyroid disease. The authors carried out their experiments using the "new thyroid" dataset from the UCI thyroid repository. Two attributes including total serum triiodothyronine (T3) and total serum thyroxin (T4) from a total of 5 attributes were used for model training. The experimental results showed that KNN outperformed other models with the highest accuracy of 96.875%. In [28], Jha et al. applied data augmentation and reduction techniques to improve the efficacy of diagnosing thyroid disease. For dimensionality reduction, the authors used Principal Component Analysis (PCA), Singular Value Decomposition (SVD), and DT; for data augmentation, they used Gaussian Distribution (GD). The dataset included 3152 samples, of which 286 samples corresponded to people with thyroid disease and 2864 samples to healthy people. The experiments demonstrated that Deep Neural Network (DNN) trained using augmented data procured a maximum accuracy of 99.95%. Yadav et al. [7] worked on the UCI repository dataset for predicting thyroid disease. Using the bagging technique, the authors proposed an ensemble of

DT, Random Forest (RF), and Extra Tree (ET). The study demonstrated that the ensemble technique produced reliable results with 100% accuracy. In [8], machine learning models—SVM, Multiple Linear Regression (MLR), and DT—were used to perform comparative thyroid disease detection, with DT achieving the highest accuracy of 97.97%.

Abbad Ur Rehman et al. [9] implemented $L_1$ and $L_2$ feature selection techniques to get effective results. The authors collected 309 patient samples from District Headquarter (DHQ) Teaching Hospital, Dera Ghazi Khan, Pakistan. The dataset comprised 10 attributes and one target column with the imbalanced distribution of three classes. Five machine learning models including KNN, SVM, DT, LR, and Naive Bayes (NB) were implemented for thyroid disease prediction. Experimental results showed that the $L_1$-based feature selection technique enhanced the performance of models. Similarly, Salman and Sonuc [29] implemented different machine learning models to determine their applicability in the detection of thyroid disease. The study included 1250 data samples from Iraqis that were collected from various hospitals and labs. The experimental results indicated that when only 13 out of the total 16 features were used for training, RF reported a maximum accuracy of 98.92%. Likewise, Shivastuti et al. [10] compared the effectiveness of SVM and RF for diagnosing thyroid disease. The research utilized 7200 data samples from a dataset available in the UCI repository. SVM surpassed RF in terms of accuracy by 1%, according to the experimental results. The experimental findings in [30] revealed that bagging ensemble integrated with three feature selection techniques including Recursive Feature Elimination (RFE), Select K-Best (SKB), and Select From Model (SFM) showed robust results in thyroid diagnosis. Similarly, Akhtar et al. [31] selected attributes from the "thyroid 0387" dataset using RFE, SKB, and SFM. The authors developed an effective unified ensemble of ensembles for improved thyroid disease diagnosis.

The aforementioned literature review is summarized in Table 1. It is worth highlighting that the majority of systems for diagnosing thyroid disease relied on attribute selection

**Table 1. A summary of recent publications for diagnosing thyroid disease using machine learning and deep learning approaches.**

| Ref | Year | Dataset | Sample Size | Target Distribution | Models | Accuracy Score |
|---|---|---|---|---|---|---|
| [26] | 2019 | Medical Facility, Kashmir | 807 | Imbalanced | KNN, DT, SVM, and NB | KNN: 91.82%, SVM: 96.52%, NB: 91.57&, and DT: 98.9% |
| [8] | 2019 | UCI | 50 | N/A | DT, SVM, and MLR | DT: 97.97% |
| [7] | 2020 | UCI | 3710 | Imbalanced | DT, RF, ET, and Ensemble | DT: 98%, RF: 99%, ET: 93%, and Ensemble: 100% |
| [27] | 2021 | UCI | 215 | Imbalanced | KNN, DT, and LR | KNN: 96.875%, DT: 87.5%, and LR: 81.25% |
| [9] | 2021 | DHQ, DG Khan, Pakistan | 309 | Imbalanced | SVM, KNN, DT, LR, and NB | KNN-$L_1$: 97.84%, KNN-$L_2$: 96.77%, DT-$L_1$: 75.34%, DT-$L_2$: 76.92%, NB-$L_1$: 100%, NB-$L_2$: 100%, SVM-$L_1$: 86.02%, SVM-$L_2$: 86.02%, LR-$L_1$: 100%, and LR-$L_2$: 98.82% |
| [29] | 2021 | Labs and Hospitals in Iraq | 1250 | N/A | DT, SVM, RF, NB, LR, KNN, LDA Linear Discriminant Analysis, and MLP Multi-layer Perceptron | DT: 98.4%, SVM: 92.27%, RF: 98.93%, NB: 81.33%, LR: 91.47%, LDA:83.2%, KNN:90.93%, and MLP: 97.6% |
| [10] | 2021 | UCI | 7200 | N/A | SVM and RF | SVM: 93% and RF: 92% |
| [30] | 2021 | DHQ, DG Khan, Pakistan | 309 | Imbalanced | RF, Base Meta Estimator (BME), AdaBoost, and XGBoost | Accuracy = 100% |
| [28] | 2022 | UCI | 3152 | Imbalanced | KNN, DNN | KNN-PCA: 94.92%, KNN-SVD: 95.72%, KNN-DT: 97.94%, DNN-PCA: 96.04%, DNN-SVD: 96.67%, DNN-DT: 98.70%, and DNN-GD: 99.95% |
| [31] | 2022 | UCI Thyroid 0387 | 7200 | N/A | RF, BME, AdaBoost, and XGBoost | LR-RFE: 99.27% |
| [35] | 2022 | UCI | 1774 | Balanced | RF, GBM, LR, AdaBoost, SVM, LSTM, CNN, CNN-LSTM | RF-MLFS: 99% |

whereas, model training was carried out using an imbalanced dataset. Numerous research demonstrated that skewed results are produced by imbalanced data [32, 33]. Nevertheless, since they lack sufficient prior knowledge, they may even provide overfitted or under-fitted predictions [34]. In [35], Chaganti et al. randomly selected 400 samples of normal class to balance the dataset comprised of 233, 346, 359, and 436 samples of primary hypothyroid, increased binding protein, compensated hypothyroid, and concurrent non-thyroidal illness classes respectively. The authors also performed attribute selection using backward, forward, and Bi-Directional elimination techniques (BFE, FFE, and BiDFE, respectively). Machine learning-based feature selection (MLFS) using ET was also performed. Several machine learning and deep learning models including RF, Gradient Boosting Machine (GBM), LR, SVM, AdaBoost, Long Short Term Memory (LSTM), Convolutional Neural Network (CNN), and CNN-LSTM were used to perform thyroid disease diagnosis. The results reported the highest accuracy of 99% by RF when integrated with attributes selected by the MLFS feature selection technique. This study serves as the basis of comparison for all of our subsequent experiments. Conversely, to balance the dataset and provide more reliable results, we adopt a downsampling approach.

## 3 Methodology

This section discusses the dataset and research methodology employed in this study for thyroid disease prediction. Initially, we acquired a dataset for thyroid disease from the UCI machine learning repository [36], which contains the data of a number of thyroid disease-related cases. Multiple target classes and samples related to thyroid disease are included in the dataset. The used dataset was highly imbalanced as the target distribution was unequal in the dataset. Our approach begins by balancing the target class distribution to produce reliable outcomes for thyroid disease detection. We used only five class samples for the experiments in which we deployed a down-sampling approach on specific data. This down-sampling approach makes the dataset balanced as discussed in Section 3.1. After data balancing, we split it into train and test sets with an 80:20 ratio. 80% of the total data was used for training of learning models, and the remaining 20% for testing purposes of the trained models. The training set was then passed to the proposed self-stacking classifier (SSC) (discussed in Section 3.2) and all other used models, which were trained for thyroid disease prediction. After training on learning models, we passed test data to perform the evaluation. For the evaluation of all models, four evaluation scores, namely accuracy, precision, recall, and F1 score were measured after receiving the test set. We also deployed 10-fold cross-validation with both the original dataset and down-sampled dataset approach and reported mean accuracy and standard deviation scores. In the end, we performed statistical analysis on the results of all approaches. For all experiments, this study used several machine learning classifiers, ensemble learners, and deep learning classifiers, followed by a comparative analysis. Fig 1 showcases the experimental workflow of this study.

All experiments were performed using a Corei7, a 12th-generation Dell machine with a Windows operating system. We used Python language to implement the proposed approach in Jupyter Notebook. Several libraries such as sci-kit learn, TensorFlow, and Keras were used for experimental purposes.

### 3.1 Data description & balancing

The data used in this study comes from the thyroid disease dataset obtained from the UCI machine learning repository [36]. The original dataset contains 31 features and 9172 sample records [35]. Table 2 presents the feature description of the underlying dataset. The target

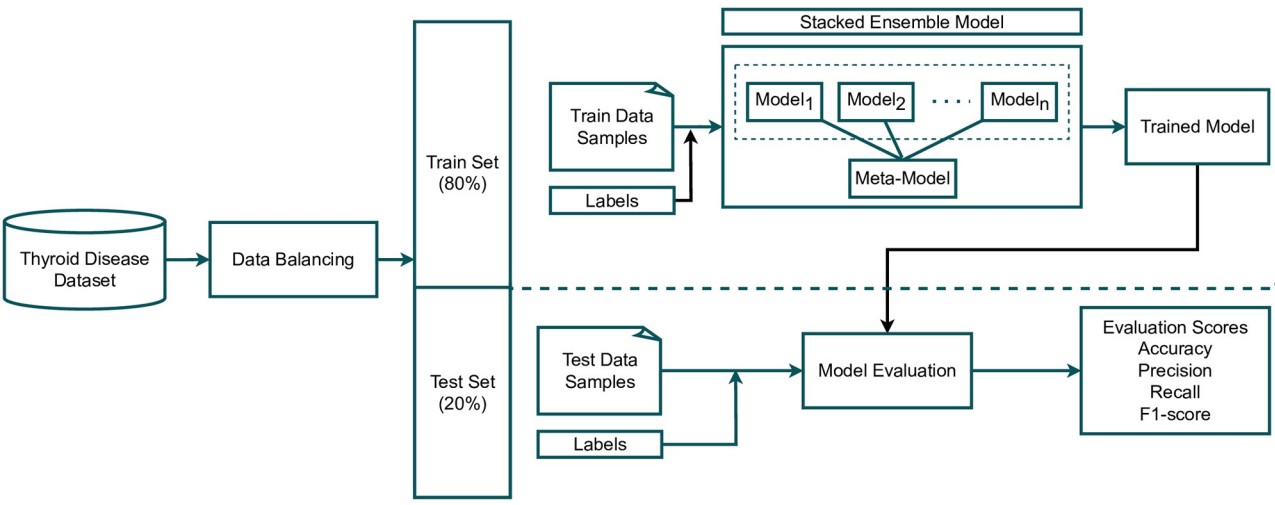

**Fig 1. Graphical abstract of this study.**

variable includes classes of thyroid disease-related health issues and diagnoses presented in Table 3. The distribution of classes is substantially imbalanced. Uneven distribution of a target variable in a dataset causes machine learning models to perform erroneously [37]. Moreover, models that have been trained on imbalanced data are ineffective at detecting minority data, and thus disease samples [38]. To address this challenge, we down-sampled the dataset and utilized the first 230 samples of each of the five diagnostic classes including primary hypothyroid (F), increased binding protein (I), compensated hypothyroid (G), concurrent non-thyroidal illness (K), and no condition (-). These target variables were selected based on the minimum sample size of 200. Table 4 displays the sample distribution of the selected target variables for the present research.

**Table 2. Feature description of thyroid disease dataset.**

| Sr. | Feature | Description | Sr. | Feature | Description |
|---|---|---|---|---|---|
| 1. | age | patients' age | 17. | TSH_measured | either patient's blood has TSH |
| 2. | sec | gender of patient | 18. | TSH | TSH level in patients' blood |
| 3. | on_thyroxine | either the person is taking thyroxine | 19. | T3_measured | either patient's blood has T3 |
| 4. | query on thyroxine | either the person is taking thyroxine | 20. | T3 | T3 level in patients' blood |
| 5. | on antithyroid meds | either the patient is taking antithyroid meds | 21. | TT4_measured | either patient's blood has TT4 |
| 6. | sick | either patient is sick or not | 22. | TT4 | TT4 level in patients' blood |
| 7. | pregnant | pregnancy status of patient | 23. | T4U_measured | either patient's blood has T4U |
| 8. | thyroid_surgery | either the patient has had thyroid surgery | 24. | T4U | T4U level in patients' blood |
| 9. | I131_treatment | whether patient is undergoing I131 treatment | 25. | FTI_measured | either patient's blood has FTI |
| 10. | query_hypo-thyroid | if the patient believes they have hypothyroidism | 26. | FTI | FTI level in patients' blood |
| 11. | query_hyperthyroid | if the patient believes they have hyperthyroidism | 27. | TGB_measured | either patient's blood has TGB |
| 12. | lithium | either the patient has lithium | 28. | TGB | TGB level in patients' blood |
| 13. | goitre | either the patient has goitre | 29. | referral_source | |
| 14. | tumor | either the patient has tumor | 30. | target | hyperthyroidism medical diagnosis |
| 15. | hypopituitary | either the patient has hyperpituitary gland | 31. | patient_id | identification number of patient |
| 16. | psych | whether patient * psych | | | |

**Table 3. Count for target variables in thyroid disease dataset.**

| Health Issues | Diagnosis | Count |
|---|---|---|
| hyperthyroid | hyperthyroid (A) | 147 |
| | T3 toxic (B) | 21 |
| | toxic goiter (C) | 6 |
| | secondary toxic (D) | 8 |
| hypothyroid | hypothyroid (E) | 1 |
| | primary hypothyroid (F) | 233 |
| | compensated hypothyroid (G) | 359 |
| | secondary hypothyroid (H) | 8 |
| binding protein | increased binding protein (I) | 346 |
| | decreased binding protein (J) | 30 |
| general health | concurrent non-thyroidal illness (K) | 436 |
| replacement therapy | under replaced (M) | 111 |
| | consistent with replacement therapy (L) | 115 |
| | over replaced (N) | 110 |
| antithyroid treatment | antithyroid drugs (O) | 14 |
| | I131 treatment (P) | 5 |
| | surgery (Q) | 14 |
| miscellaneous | discordant assay results (R) | 196 |
| | elevated TBG (S) | 85 |
| | elevated thyroid hormones (T) | 0 |
| no condition | sample record of healthy person (-) | 6771 |

**Table 4. Down-sampled data for efficient thyroid disease detection.**

| Target Variable | Original Sample Count | Down-sampled Count |
|---|---|---|
| primary hypothyroid (F) | 233 | 230 |
| increased binding protein (I) | 346 | 230 |
| compensated hypothyroid (G) | 359 | 230 |
| concurrent non-thyroidal illness (K) | 436 | 230 |
| no condition (-) | 6771 | 230 |

Figs 2 & 3, the feature importance scores are displayed for both the original dataset and the down-sampled dataset. These scores were obtained using the Extra Trees Classifier (ETC), which takes all features and the target into account for both scenarios. The ETC is a tree-based classifier that determines the importance of each feature by calculating its entropy criterion. The feature importance scores for both scenarios are highlighted in Figs 2 and 3. It is observed that there is only a minor difference between the feature importance scores for both scenarios and the range of feature importance scores has increased after balancing the data which helps to improve the performance of learning models in this study.

## 3.2 Proposed Self Stacking Classifier (SSC)

This study proposes a novel stacked ensemble, called a self-stacking classifier (SSC). The traditional stacking model combines heterogeneous estimators (base learners) to minimize their errors and then provide them as inputs to the final estimator (meta-learner). The stacked classifier leverages the power of multiple base learners on a classification problem to create

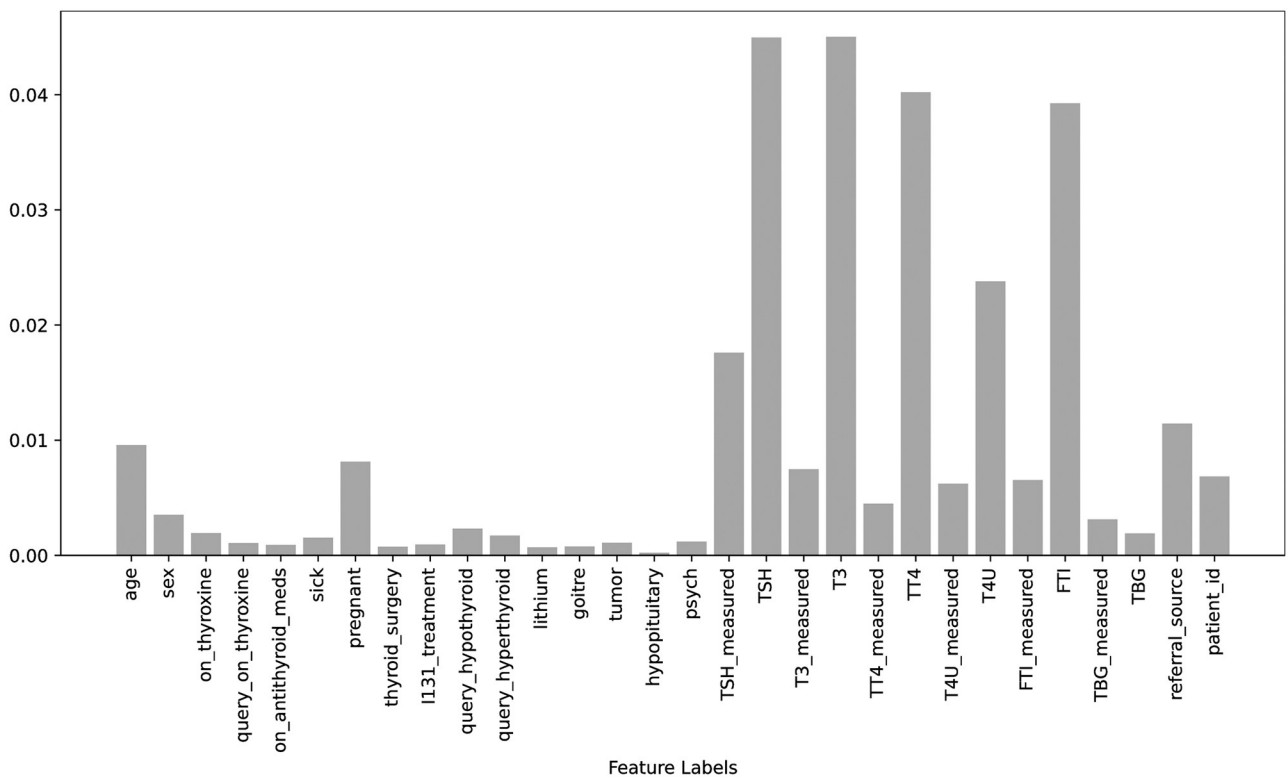

**Fig 2. Features importance scores using original dataset.**

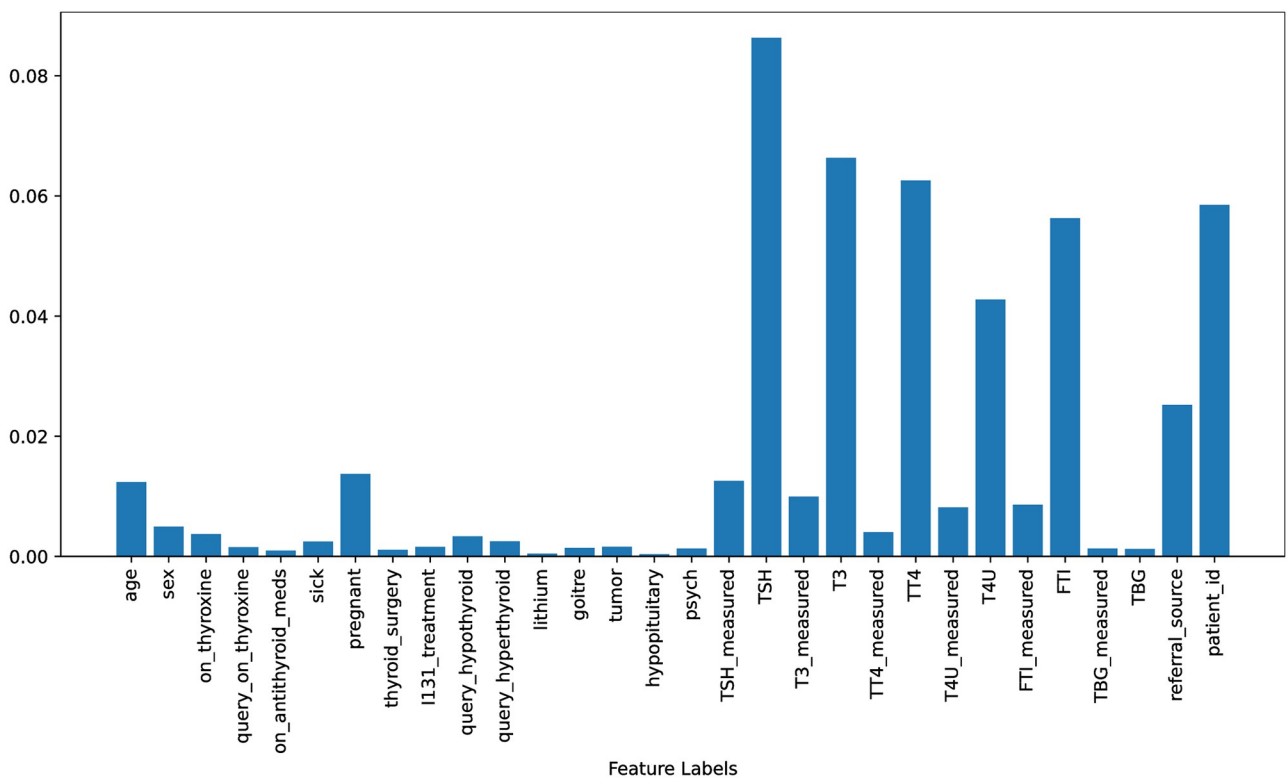

**Fig 3. Features importance scores using down-sampled dataset.**

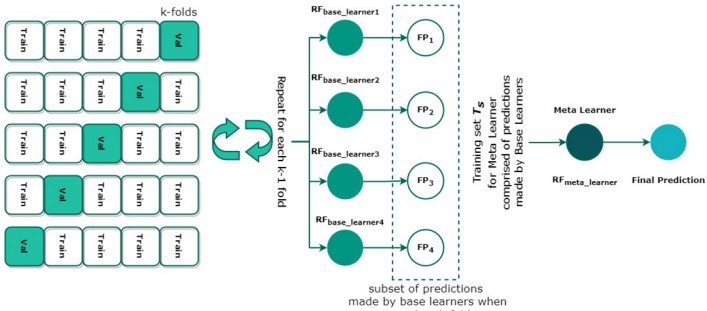

**Fig 4. Architecture of proposed Self Stacking Classifier comprised of RF as a base and meta learner.**

predictions. These output predictions act as features for the training of meta-classifiers. Finally, the meta-stacking technique is used to find the optimum way to integrate predictions from several underlying base learners to give the final output. Fig 4 shows the architecture of SSC.

Contrary to heterogeneous base learners and meta-learners, this study adopts a self-stacking ensemble where the same machine learning model i.e. RF, serves as a base learner. We selected RF—a combination of various de-correlated decision trees (DTs)—because of its high efficacy in high-dimensional classification problems [39]. Consider a training set $T = \{X_i, Y_i\}$ where $i = 1, 2, 3, \ldots, n$ is the number of vectors, $X \in S$ where $S$ is a set of sample data and $Y \in C$ where $C$ is a set of target variables. When applied to the underlying problem, the model maps the input sample data to the target variable $S \rightarrow C$. Each DT in RF classifies a new input vector yielding a certain output prediction. RF builds DTs by incorporating training bootstrapped samples from $T$ and selecting a subset of $S$ at each node. This method of training is used for each $RF_{base\_learner_m}$ where $m = 1, \ldots, 4$, functioning as a base learner. Moreover, the stacking model integrates k-fold cross-validation for the training of each $RF_{base\_learner_m}$ in the proposed SSC model. The final predictions $FP_k$ made by each fold of the k-fold act as features for the training of meta-classifier ($RF_{meta\_learner}$). Finally, $RF_{meta\_learner}$ determines the optimal combination of $FP_m$ to finally classify the underlying data sample and detect the underlying thyroid disease. Algorithm 1 presents the pseudocode of the proposed SSC model.

**Algorithm 1** Pseudocode of proposed Self-Stacking Classifier

```
Input: Training Set, T = {Sᵢ, Cᵢ} where, Sᵢ=input features and Cᵢ=tar-
get variable, i = 1 to 4 is the number of base learners.
Output: Self-Stacking Classifier X_SSC
1: Step 1: Learn about RF_base_learnersᵢ where, i = 1 to 4
2:   for t ← 1 to 4 do
3:     Learn RF_base_learnersₜ based on T
4:   end for
5: Step 2: Construct new training dataset Tₛ from T
6:   for i ← 1 to m do
7:     Tₛ = {S'ᵢ, Cᵢ} where, S'ᵢ = {RF_base_learners₁(Sᵢ),...,RF_base_learnersₙ(Sᵢ)}
8:   end for
9: Step 3: Learn the meta classifier RF_meta_learner
10:    Train RF_meta_learner on newly constructed dataset Tₛ
11: return X_SSC(S) = RF_meta_learner(RF_base_learner₁(S),...RF_base_learner₄(S))
```

## 3.3 Algorithms tested

In this study, we tested several machine learning algorithms which have significant applications in different domains, such as the health care [40], Internet of Things (IoT) [41], machine

**Table 5. Hyperparameter settings for machine learning algorithms.**

| Algorithm | Hyperparameter | Tuning Range |
|---|---|---|
| RF | n_estimators = 200, max_depth = 50 | n_estimators={20 to 200}, max_depth={5 to 100} |
| GBM | max_depth = 200, learning_rate = 0.2, n_estimators = 50, random_state = 52 | n_estimators={20 to 200}, max_depth={5 to 100}, random_state={2 to 100 }, learning_rate= {0.1 to 0.9 } |
| AdaBoost | n_estimators = 300, random_state = 5, learning_rate = 0.8 | n_estimators={20 to 200}, random_state={2 to 100 }, learning_rate= {0.1 to 0.9 } |
| LR | solver='saga', multi_class='multinomial', C = 3.0 | solver= ['saga','sag','liblinear'], multi_class='multinomial', C={1.0 to 5.0} |
| SVC | kernel='linear', C = 1.0 | kernel=['linear', 'poly', 'sigmoid'], C={1.0 to 5.0} |

vision [42], edge computing [43], education [44, 45], and many others. In order to conduct a fair comparative evaluation of our proposed SSC model for the detection of thyroid disease, we chose the following machine learning classifiers: RF due to its effectiveness, interpretability, non-parametric nature, and high accuracy rate across a range of data types; GBM, which has various benefits including adaptability, robust tolerance to anomalous inputs, and high accuracy; AdaBoost, since it is less susceptible to overfitting; LR, because its training and implementation processes are simple; and Support Vector Classifier (SVC), that has advantages including efficiently handling high dimensional data [46]. For these algorithms to perform at their maximum, we optimized their hyperparameters. We select the best hyperparameter setting by tuning the models on specific value ranges. Table 5 presents the hyperparameter settings for each algorithm.

Likewise, we utilized self voting ensemble classifier (SVEC) to further emphasize the significance and usefulness of the proposed classifier. The traditional voting classifier is a meta-classifier for classification based on "hard" or "soft" voting criteria that integrate similar or distinct machine learning models. Contrary to this, we adopted a self-voting mechanism in which the output of the same model is combined to produce the final prediction. We employed RF as the base learner in SVEC and used both hard and soft voting criteria to classify the thyroid disease dataset. Hard voting adheres to the basic majority voting criteria. Soft voting classifies target values by aggregating the projected probabilities of the base classifiers and selecting the target variable with the highest probability. Algorithms 2 and 3 demonstrate the pseudocodes for hard voting (SVEC-H) and soft voting (SVEC-S) respectively. The mode in SVC-H Algorithm 2, represents that the target class with the highest vote will be the final prediction, and argmax in SVC-S Algorithms 3, represent that the class with the highest average probability will be the final prediction.

**Algorithm 2** Pseudocode of *Hard* Self-Voting Ensemble Classifier

```
Input
T = Training set representing A target variables.
Classifier_base = Base Classifiers i.e. RF_base1, RF_base2, and RF_base3
L = Labels of training data.
D = Number of base classifiers.
Output: Diagnostic Class
1: procedure:
2: do n = 1 to D
3:    Call Classifier_base with T_n
4:    Receive the classifier Classifier_base_n
5:    Compare L_n with A_n generated by Classifier_base_n
6:    Update vote V
7: Final Prediction← mode{RF_base1(V), RF_base2(V), RF_base3(V) }
```

**Algorithm 3** Pseudocode of *Soft* Self-Voting Ensemble Classifier

```
Input:
```

```
T = Training set
D = Testing Set
Classifier_base = Base Classifiers i.e. RF_base1, RF_base2, and RF_base3
Output: Diagnostic Class
1: Step 1: Training base classifiers
2:    R_1 = RF_base1(T)
3:    R_2 = RF_base2(T)
4:    R_3 = RF_base3(T)
5:    Step 2: Testing base classifiers
6:    P(R_1) ← D
      P(R_1) = {a_1, a_2, ..., a_n}
7:    P(R_2) ← D
      P(R_2) = {b_1, b_2, ..., b_n}
8:    P(R_3) ← D
      P(R_3) = {c_1, c_2, ..., c_n}
9:    Final Prediction ← argmax(∑_{i=0}^{n}{P(R_1) + P(R_2) + P(R_3)})
```

To further substantiate the performance of the SSC classifier, deep neural networks were also taken into consideration. A deep neural network consists of artificial neurons that have been connected and exchanged their output with neighboring neurons. They incorporate an optimization or loss function to maximize the output in conjunction with input, hidden, and output layers. For the intended result, the weights are increased in every subsequent epoch. We employed CNN and LSTM, two well-known deep neural networks. The hyperparameter setting for deep learning models is according to the literature. We study the researchers who worked on the same kinds of datasets and we used the same kind of state-of-the-art architectures to achieve significant accuracy [35, 44, 47]. Additionally, we integrated these neural networks and used a CNN-LSTM hybrid to detect thyroid disease. Table 6 provides the layered architecture of the deep neural networks used in this study.

## 3.4 Performance estimation evaluation estimators

The performance of classifiers used in this study is assessed using accuracy and macro estimators (macro precision, macro recall, and macro F1-score). The rationale for using macro estimators is that the micro estimators aggregate the average parameter and produce results that are biased toward classes with many instances. Likewise, weighted estimators consider the sample size for each target variable when computing the average, resulting in skewed results in the case of imbalanced data.

The very unbalanced nature of the dataset utilized in this work necessitated the adoption of macro estimators, which allowed us to fairly compare classifier performance on unbalanced data with that on balanced data. The corresponding formulas to compute accuracy, precision,

**Table 6. Layered architecture of deep neural networks.**

| LSTM | CNN | CNN-LSTM |
|---|---|---|
| Embedding(5000,200) | Embedding(5000,200) | Embedding(5000,200) |
| Dropout(0.5) | Conv1D(128,5,activation='relu') | Conv1D(128,5,activation='relu') |
| LSTM(128) | MaxPooling1D(pool_size = 5) | MaxPooling1D(pool_size = 2) |
| Dropout(0.5) | Activation('relu') | LSTM(64,return_sequences = True) |
| LSTM(64) | Dropout(rate = 0.5) | Conv1D(128,5,activation='relu') |
| Dropout(0.5) | Flatten() | MaxPooling1D(pool_size = 2) |
| Dense(32) | Dense(32) | Flatten() |
| Dense(5) | Dense(5) | Dense(5) |
| loss ='categorical_crossentropy', optimizer ='adam', epochs = 100, batch_size = 128, activation='softmax' | | |

recall, and F1-score are listed below. Albeit the weighted, micro, and macro estimators are all calculated in a different manner, however, these estimators compute precision, recall, and F1-score using the same formulas.

$$Accuracy = \frac{TP + TN}{TP + FP + TN + FN} \tag{1}$$

$$Precision = \frac{TP}{TP + FP} \tag{2}$$

$$Recall = \frac{TP}{TP + FN} \tag{3}$$

$$F1 - score = 2 \times \frac{precision \times recall}{precision + recall} \tag{4}$$

where TP (True Positive) and TN (True Negative) are the numbers of correctly predicted positive and negative instances respectively. FP (False Positive) is the count of negative instances that were predicted as positive instances. Similarly, FN (False Negative) is the number of positive instances incorrectly predicted as negative instances. All evaluation metrics are standardized with values ranging from 0 to 1, where 0 represents the lowest score and 1 represents the highest score.

## 4 Results

We performed two distinct sets of experiments. We were initially, using the original distribution of data samples across each target variable, and later on adopted down-sampled data for model training. The training and test sample distribution for each case is shown in Fig 5.

### 4.1 Experimental results of machine learning classifiers

First, experiments were carried out on original and down-sampled datasets using individual machine-learning classifiers. In Table 7, we can observe that the accuracy (the degree to which the classifier delivered correct predictions) of RF is the highest among all classifiers in both settings for thyroid detection using machine learning classifiers. The model yielded 98% and 97% accuracy when experimented with original and down-sampled datasets respectively. However, it is important to draw attention to the significant difference in the classifiers' accuracy, precision, recall, and F1 scores when trained and tested using the original distribution.

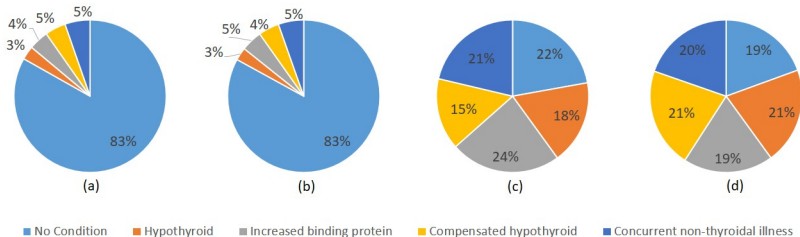

**Fig 5.** Distribution of original and down-sampled train and test samples across each target class (a) Training set of original data, (b) Test set of original data, (c) Training set of down-sampled data, and (d) Test set of down-sampled data.

**Table 7. Experimental results of machine learning classifiers.**

| Classifier | Original Data | | | | Down-Sampled Data | | | |
|---|---|---|---|---|---|---|---|---|
| | Accuracy | Precision | Recall | F1 | Accuracy | Precision | Recall | F1 |
| RF | 0.98 | 0.97 | 0.94 | 0.95 | 0.97 | 0.98 | 0.97 | 0.97 |
| GBM | 0.97 | 0.92 | 0.94 | 0.93 | 0.95 | 0.95 | 0.95 | 0.95 |
| AdaBoost | 0.88 | 0.72 | 0.69 | 0.66 | 0.94 | 0.94 | 0.94 | 0.94 |
| LR | 0.86 | 0.52 | 0.40 | 0.43 | 0.89 | 0.88 | 0.88 | 0.88 |
| SVC | 0.83 | 0.17 | 0.20 | 0.18 | 0.93 | 0.93 | 0.92 | 0.92 |

Nevertheless, with a balanced distribution of data, RF produced more stable results despite achieving higher accuracy on the original dataset. The remaining classifiers, including GBM, AdaBoost, and SVC, obtained accuracy above 92% for balanced data. Hence, it is viable to discern that classifiers perform better overall when the dataset has been balanced. The linear model LR, on the contrary, presented poor performance.

Table 8 shows the target class-wise results of machine learning classifiers. It is evident that the classifiers performed poorly for minority classes, such as classes with few training instances, when the target classes' original distribution, or imbalanced data, was used. In comparison, more consistent results can be seen when the dataset is distributed evenly, or when

**Table 8. Target class-wise results of machine learning classifiers.**

| Classifier | Target Class | Original Data | | | Down-Sampled Data | | |
|---|---|---|---|---|---|---|---|
| | | Precision | Recall | F1 | Precision | Recall | F1 |
| RF | No Condition | 0.98 | 0.99 | 0.99 | 0.91 | 1.00 | 0.95 |
| | Hypothyroid | 0.98 | 0.95 | 0.97 | 0.98 | 1.00 | 0.99 |
| | Increased binding protein | 0.96 | 0.99 | 0.97 | 1.00 | 1.00 | 1.00 |
| | Compensated hypothyroid | 0.96 | 0.76 | 0.85 | 1.00 | 0.86 | 0.92 |
| | Concurrent non-thyroidal illness | 0.98 | 0.99 | 0.98 | 1.00 | 0.98 | 0.99 |
| GBM | No Condition | 0.99 | 0.98 | 0.98 | 0.89 | 0.96 | 0.92 |
| | Hypothyroid | 0.98 | 0.91 | 0.94 | 1.00 | 1.00 | 1.00 |
| | Increased binding protein | 0.92 | 0.99 | 0.95 | 0.98 | 1.00 | 0.99 |
| | Compensated hypothyroid | 0.78 | 0.82 | 0.80 | 0.91 | 0.83 | 0.87 |
| | Concurrent non-thyroidal illness | 0.95 | 0.99 | 0.97 | 0.98 | 0.94 | 0.96 |
| AdaBoost | No Condition | 0.92 | 0.94 | 0.93 | 0.90 | 0.92 | 0.91 |
| | Hypothyroid | 1.00 | 0.61 | 0.76 | 1.00 | 1.00 | 1.00 |
| | Increased binding protein | 0.93 | 0.99 | 0.96 | 0.98 | 1.00 | 0.99 |
| | Compensated hypothyroid | 0.43 | 0.85 | 0.57 | 0.79 | 0.89 | 0.84 |
| | Concurrent non-thyroidal illness | 0.33 | 0.05 | 0.08 | 1.00 | 0.88 | 0.93 |
| LR | No Condition | 0.86 | 0.99 | 0.92 | 0.87 | 0.92 | 0.90 |
| | Hypothyroid | 0.87 | 0.77 | 0.82 | 0.90 | 0.93 | 0.92 |
| | Increased binding protein | 0.00 | 0.00 | 0.00 | 0.94 | 0.85 | 0.89 |
| | Compensated hypothyroid | 0.86 | 0.25 | 0.39 | 0.82 | 0.80 | 0.81 |
| | Concurrent non-thyroidal illness | 0.00 | 0.00 | 0.00 | 0.88 | 0.92 | 0.90 |
| SVC | No Condition | 0.83 | 0.99 | 0.91 | 0.85 | 1.00 | 0.92 |
| | Hypothyroid | 0.00 | 0.00 | 0.00 | 0.91 | 0.95 | 0.93 |
| | Increased binding protein | 0.00 | 0.00 | 0.00 | 0.96 | 0.87 | 0.91 |
| | Compensated hypothyroid | 0.00 | 0.00 | 0.00 | 0.94 | 0.86 | 0.90 |
| | Concurrent non-thyroidal illness | 0.00 | 0.00 | 0.00 | 1.00 | 0.94 | 0.97 |

**Table 9. Experimental results of ensemble voting classifiers.**

| Classifier | Original Data | | | | Down-Sampled Data | | | |
|---|---|---|---|---|---|---|---|---|
| | Accuracy | Precision | Recall | F1 | Accuracy | Precision | Recall | F1 |
| SVEC-H | 0.98 | 0.96 | 0.94 | 0.95 | 0.97 | 0.98 | 0.96 | 0.96 |
| SVEC-S | 0.98 | 0.96 | 0.94 | 0.95 | 0.97 | 0.98 | 0.96 | 0.96 |

**Table 10. Target class-wise classification results of self-voting ensemble classifiers.**

| Classifier | Target Class | Original Data | | | Down-Sampled Data | | |
|---|---|---|---|---|---|---|---|
| | | Precision | Recall | F1 | Precision | Recall | F1 |
| SVEC-H | No Condition | 0.99 | 0.99 | 0.99 | 0.89 | 1.00 | 0.94 |
| | Hypothyroid | 0.98 | 1.00 | 0.99 | 1.00 | 1.00 | 1.00 |
| | Increased binding protein | 0.96 | 1.00 | 0.98 | 0.98 | 1.00 | 0.99 |
| | Compensated hypothyroid | 0.96 | 0.71 | 0.82 | 1.00 | 0.80 | 0.89 |
| | Concurrent non-thyroidal illness | 0.92 | 0.98 | 0.95 | 1.00 | 1.00 | 1.00 |
| SVEC-S | No Condition | 0.99 | 0.99 | 0.99 | 0.89 | 1.00 | 0.94 |
| | Hypothyroid | 0.98 | 1.00 | 0.99 | 1.00 | 1.00 | 1.00 |
| | Increased binding protein | 0.96 | 1.00 | 0.98 | 0.98 | 1.00 | 0.99 |
| | Compensated hypothyroid | 0.96 | 0.71 | 0.82 | 1.00 | 0.80 | 0.89 |
| | Concurrent non-thyroidal illness | 0.92 | 0.98 | 0.95 | 1.00 | 1.00 | 1.00 |

the data are down-sampled. Because classifiers were able to learn and interpret the features of each class in a balanced manner, the results were more reliable.

## 4.2 Experimental results of self-voting classifiers

In this study, we also implement and assess the performance of ensemble classification under self-voting criteria. A voting classifier, as described in Section 3.3, is a collection of various machine learning classifiers that perform prediction using either soft or hard procedures. We ran experiments using the developed self-voting SVEC-H and SVEC-S ensembles of three RF classifiers under hard and soft voting criteria, respectively, to investigate the applicability of self-voting classifiers on the underlying dataset. The experimental results obtained by self-voting classifiers are shown in Table 9. Both self-voting classifiers achieved accuracy values greater than 97%, as shown. The uneven distribution of target classes, however, leads to inconsistent results. Additionally, similar percentages of performance estimators could be seen for both hard and soft voting criteria. Albeit the RF classifier individually achieved 97% accuracy with down-sampled data, it is clear that the self-voting classifier achieves a comparable 97% accuracy score on down-sampled data. Moreover, it is apparent from Table 10 that the classification results of the majority voting (hard) and weighted probability voting (soft) criterion remained similar for each target class.

## 4.3 Experimental results of deep neural networks

We also evaluated the performance of deep neural networks for thyroid disease detection using the underlying original and balanced distribution of the dataset. Table 11 presents the classification results of deep neural networks. CNN yielded the highest accuracy score of 95% when trained and tested using the original distribution of the dataset. A significant drop in the evaluation estimators can be observed with a balanced dataset distribution. Another pertinent

**Table 11. Deep learning models results for thyroid disease classification.**

| Classifier | Original Data | | | | Down-Sampled Data | | | |
|---|---|---|---|---|---|---|---|---|
| | Accuracy | Precision | Recall | F1 | Accuracy | Precision | Recall | F1 |
| LSTM | 0.92 | 0.76 | 0.73 | 0.73 | 0.84 | 0.83 | 0.84 | 0.83 |
| CNN | 0.95 | 0.89 | 0.81 | 0.84 | 0.90 | 0.91 | 0.90 | 0.90 |
| CNN-LSTM | 0.93 | 0.84 | 0.76 | 0.79 | 0.79 | 0.79 | 0.79 | 0.79 |

point is the inconsistency between the evaluation estimators such as accuracy precision, recall, and F1-score when the deep neural networks dealt with imbalanced data.

On the contrary, the models yielded lower but consistent and reliable results with balanced data. Fig 6 showcases the performance estimation of deep neural networks with original and downs-sampled data across each epoch. A significant decline in the performance of CNN-LSTM can be observed. Furthermore, CNN demonstrated rather more stability and comparatively improved performance in the classification of target values with each successive epoch. The lower performance results of evaluation estimators are because the deep learning models have the tendency to work better with larger datasets. As the number of data instances reduced with the down-sampling technique, the performance of deep neural networks also degraded. The terms 'Accuracy' and 'Val_Accuracy' in Figs 6 and 7 correspond to the accuracy metrics computed on the training and validation datasets, respectively. This same concept extends to other evaluation metrics, where 'val_precision' represents validation precision, 'val_recall' stands for validation recall, and 'val_F1_Score' stands for the validation F1 score.

Fig 7 displays the loss of information across each epoch. When balanced data—that is, down-sampled data—was given to deep neural networks, the validation loss decreased overall,

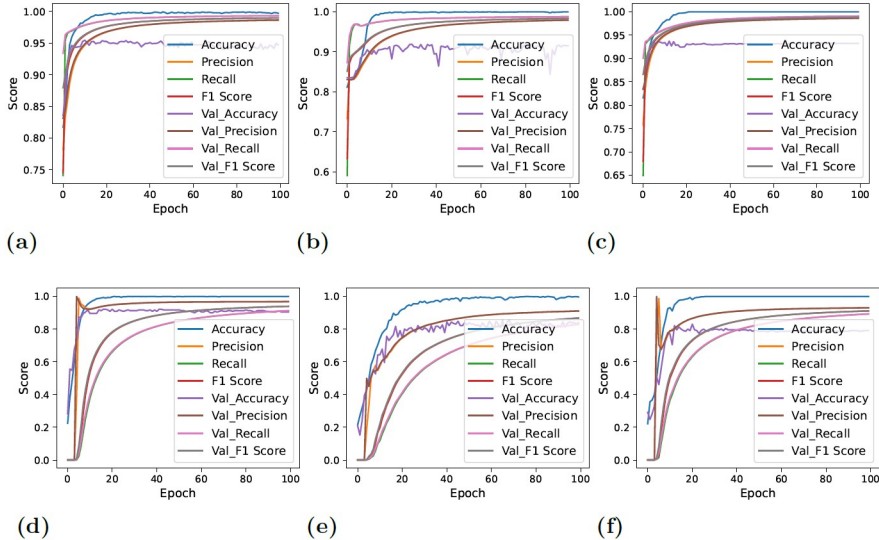

**Fig 6. Per epochs evaluation score for deep learning models.** In these figures, 'val' indicates validation, so 'val_accuracy' represents the accuracy on the validation set, and the same principle applies to other metrics (a) CNN + Original Data, (b) LSTM + Original Data, (c) CNN-LSTM + Original Data, (d) CNN + down-sampled data, (e) LSTM + down-sampled data, and (f) CNN-LSTM + down-sampled data.

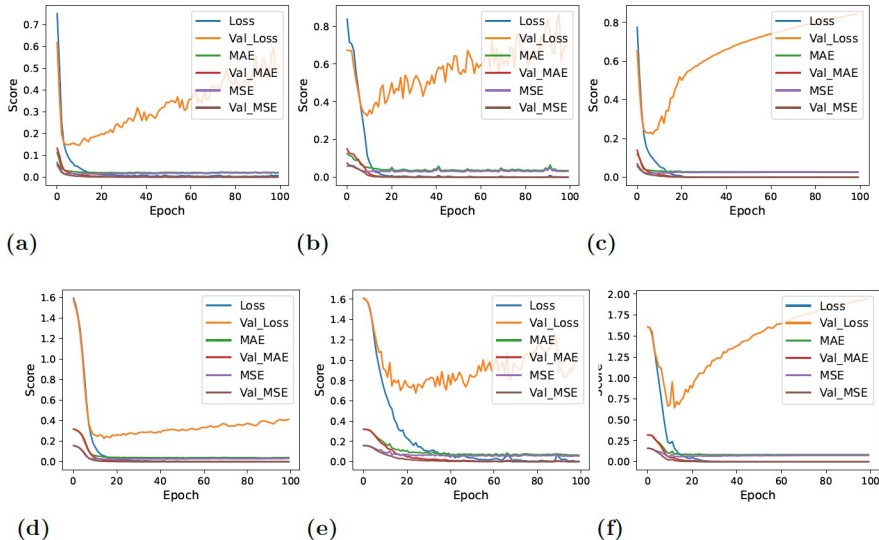

**Fig 7. Per epochs loss estimation for deep learning models.** In these figures, 'val' indicates validation, so 'val_loss' represents the loss on the validation set, and the same principle applies to other metrics (a) CNN + Original Data, (b) LSTM + Original Data, (c) CNN-LSTM + Original Data, (d) CNN + down-sampled data, (e) LSTM + down-sampled data, and (f) CNN-LSTM + down-sampled data.

as can be shown. The CNN-LSTM loss graph, however, continued to be the highest, demonstrating the model's poor performance in detecting thyroid disease.

## 4.4 Experimental results of proposed self-stacking classifier

After experimental verification, it was shown that RF was the superlative classifier for the thyroid disease classification dataset. However, it was believed that more enhanced and reliable results were required. Besides that, as described in Section 4.2, the self-voting technique was found to be comparatively inefficient for identifying thyroid disease. As a result, we explored further into the performance of stacking, another ensemble technique. To do so, we suggested and put into practice a self-stacking classifier (SSC) based on RF. Even though it performed relatively poorly on the original dataset, the proposed SSC model displayed state-of-the-art

**Table 12. Classification results of proposed SSC model.**

| Data Distribution | Accuracy | Target | Precision | Recall | F1 |
|---|---|---|---|---|---|
| Original Data | 0.98 | No Condition | 0.99 | 0.99 | 0.99 |
| | | Hypothyroid | 0.98 | 0.93 | 0.95 |
| | | Increased binding protein | 0.96 | 0.99 | 0.97 |
| | | Compensated hypothyroid | 0.90 | 0.85 | 0.87 |
| | | Concurrent non-thyroidal illness | 0.96 | 0.99 | 0.97 |
| | | Macro Average | 0.95 | 0.95 | 0.95 |
| Down-Sampled Data | 0.995 | No Condition | 1.00 | 0.98 | 0.99 |
| | | Hypothyroid | 1.00 | 1.00 | 1.00 |
| | | Increased binding protein | 1.00 | 1.00 | 1.00 |
| | | Compensated hypothyroid | 1.00 | 1.00 | 1.00 |
| | | Concurrent non-thyroidal illness | 0.98 | 1.00 | 0.99 |
| | | Macro Average | 1.00 | 1.00 | 1.00 |

performance when supplied with balanced data, as demonstrated in Table 12. In terms of diagnosing thyroid disease, the model scored a state-of-the-art 99.5% accuracy score. Other evaluation estimators including precision, recall, and F1-score all showed a perfect score of 100% demonstrating the reliability and effectiveness of the proposed model. Additionally, the SSC model successfully classified each class with the highest degrees of precision, recall, and F1 scores. This also contributes to the robustness of the proposed model.

## 5 Comparative analysis of experimental results

In this section, we compare and discuss the extensive experimentation results to further demonstrate the robustness of the proposed SSC model. As evaluation criteria for the comparative study, we used confusion matrices. The main diagonal of the confusion matrix displays the number of correctly predicted instances in correspondence to each associated class. In the confusion matrix the labels "0,1,2,3, and 4" indicate "no condition", hypothyroid, increased

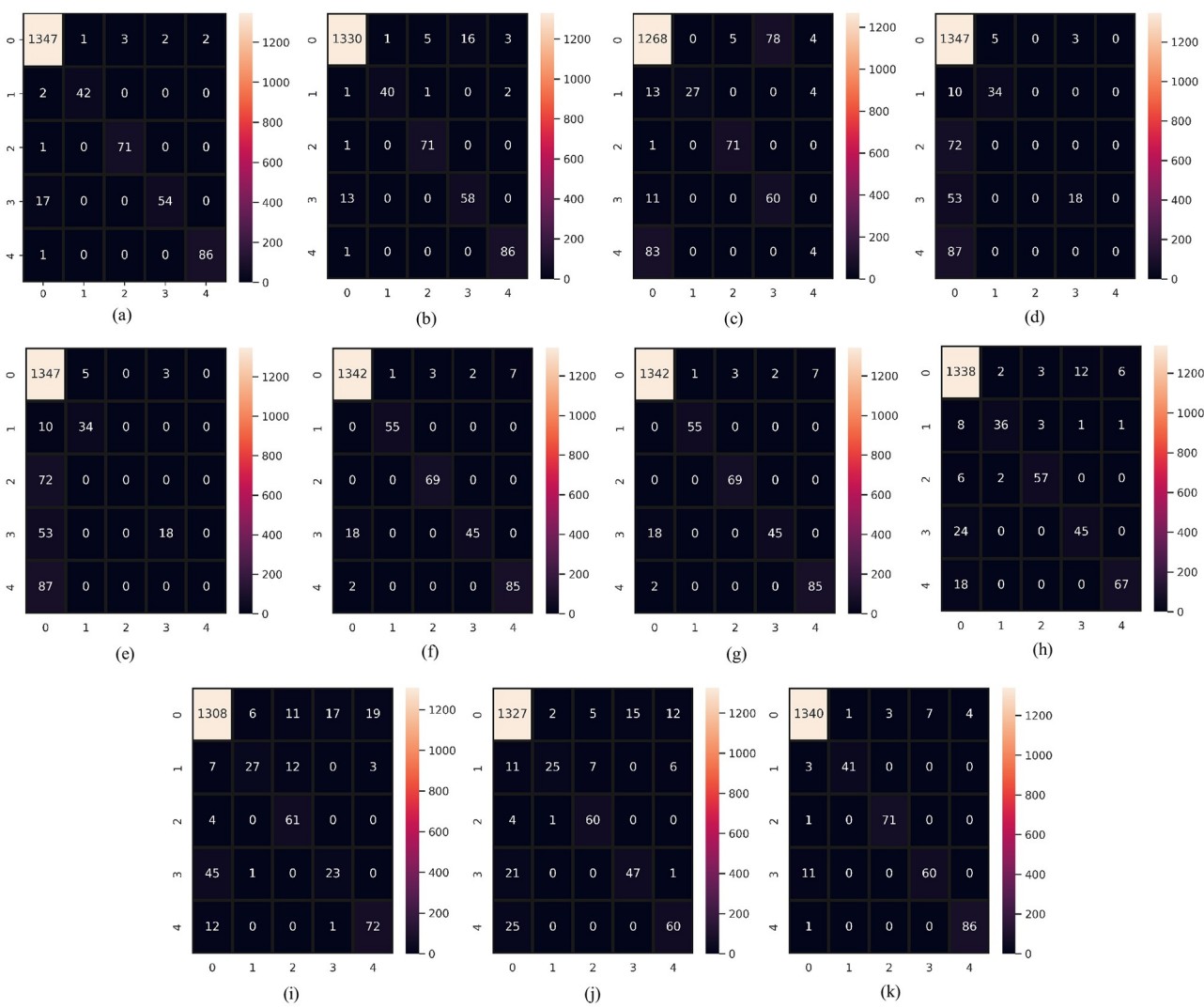

**Fig 8. Confusion matrix of classifiers when supplied with original data.** (a) RF, (b) GBM, (c) AdaBoost, (d) LR, (e) SVC, (f) SVEC-H, (g) SVEC-S, (h) CNN, (i) LSTM, (j) CNN-LSTM, and (k) SSC.

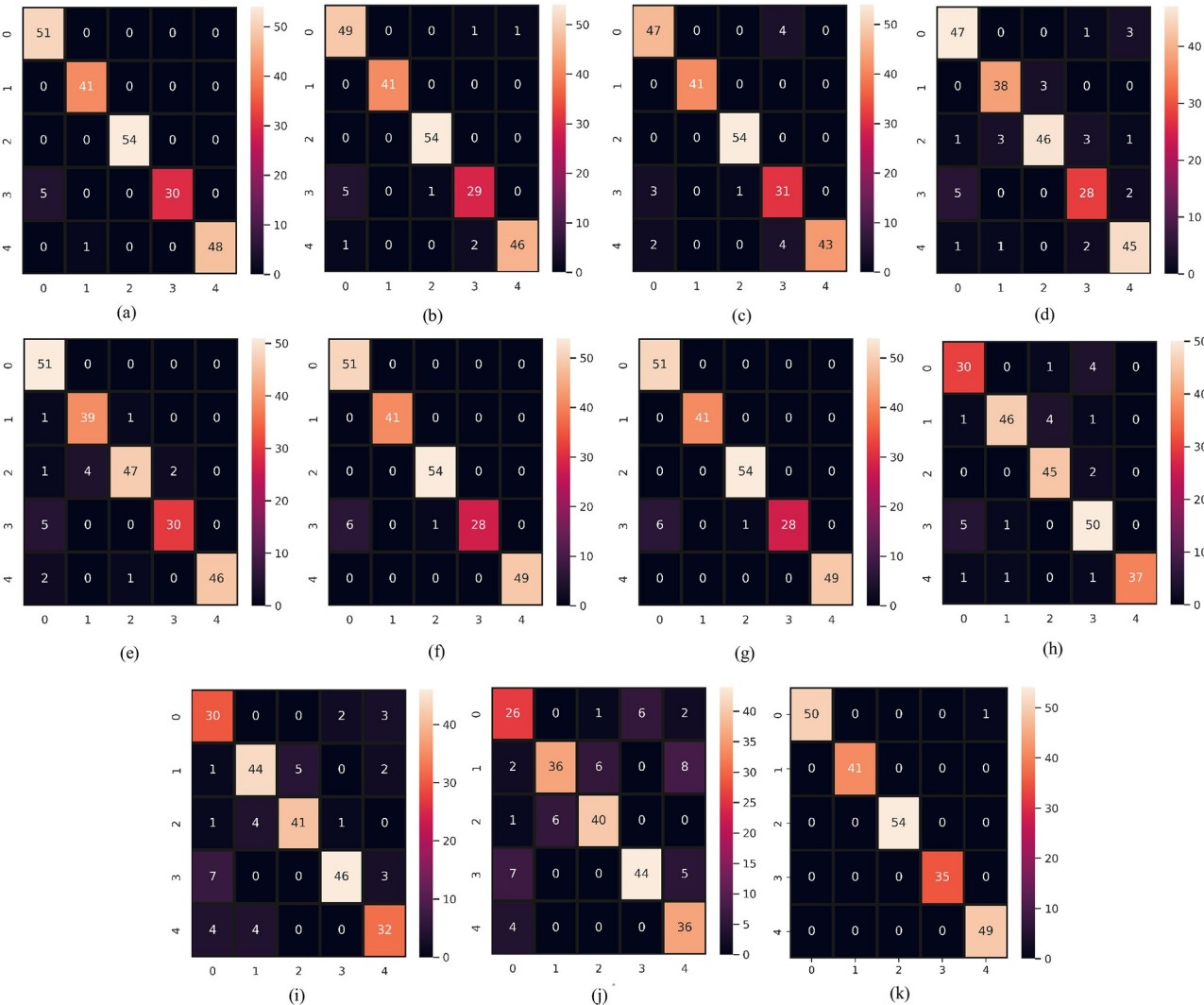

**Fig 9. Confusion matrix of classifiers when supplied with down-sampled data.** (a) RF, (b) GBM, (c) AdaBoost, (d) LR, (e) SVC, (f) SVEC-H, (g) SVEC-S, (h) CNN, (i) LSTM, (j) CNN-LSTM, and (k) SSC.

binding protein, compensated hypothyroid, and concurrent non-thyroidal illness classes respectively. Figs 8 and 9 show the confusion matrix for both machine learning and deep learning models on the same size of test data. We have done separate data splitting (Training and Testing sets) for machine learning and deep learning models. When splitting the dataset, we used the shuffle parameter which changes the number of instances for each target class for both machine and deep learning models but in both cases, the total number of test samples is the same which is 230. We did splitting separately for deep learning models because the last layer of deep learning has 5 neurons. For that, we have to feed target data after converting it into five variables and we split data again after the conversion. We use shuffle parameters in the train test split to change the count for each category in the test dataset.

The "no condition" target class makes up over 80% of the original dataset; in contrast, thyroid disease classes make up only 20% of the remaining dataset, making them the minority class. The models have the tendency to over-fit on the majority class consequently, performing

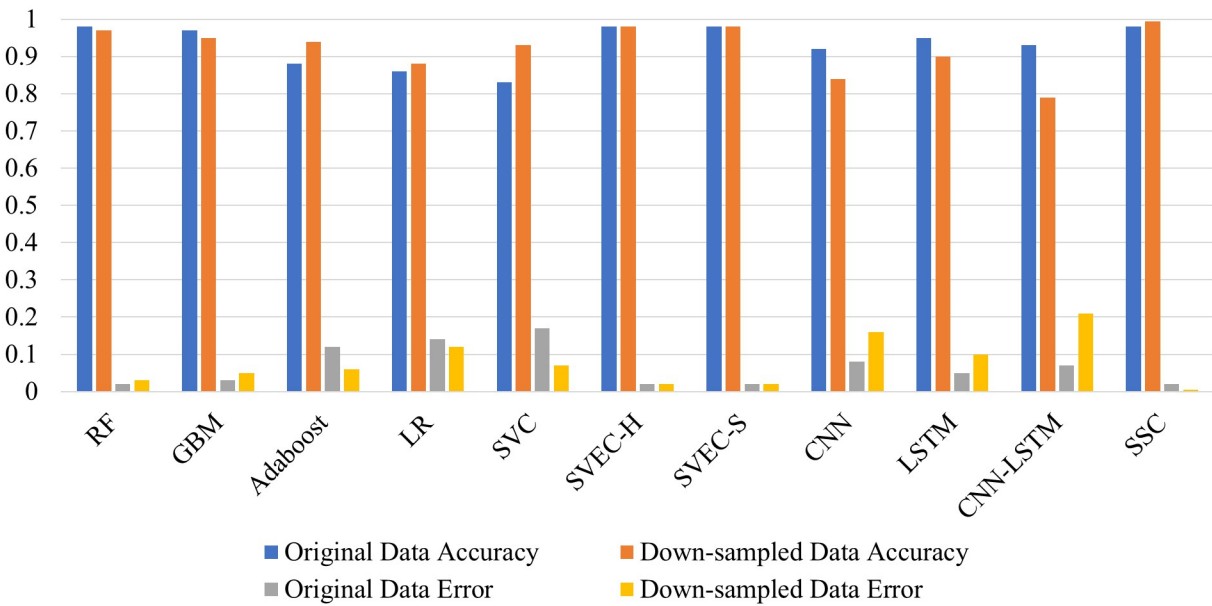

**Fig 10. Performance analysis of classifiers in terms of accuracy and error rate.**

poorly on minority classes. The visual representation of the classifier's performance on imbalanced data is shown in Fig 8 through their respective confusion matrix. In contrast to the minority classes, it is evident that the class with the majority of the training samples was classified correctly. The model's RF and SSC produced the lowest percentage of test instances that were erroneously predicted, with RF maintaining the highest percentage in the case of unbalanced data with only 29 inaccurate predictions. However, SSC delivered 33 incorrect predictions. Despite this, each classifier experienced a decline in the number of incorrect predictions with the balanced dataset, with the proposed SSC coming out on top with only one incorrect prediction. Moreover, classifiers recognized the minority samples more correctly. This shows the efficacy of the proposed model. Fig 10 presents a bar chart comparing the classifier's performance in identifying thyroid disease from an imbalanced and balanced dataset. This also displays the superior performance of the proposed SSC when provided with balanced data.

We employed various methods such as 10-fold cross-validation, standard deviation (SD), and time measurement to further compare, validate, and generalize the performance of the

**Table 13. 10-fold cross-validation results of classifiers.**

| Classifier | Original Data | | | Down-Sampled Data | | |
|---|---|---|---|---|---|---|
| | Accuracy | SD | Time (sec) | Accuracy | SD | Time (sec) |
| RF | 0.97 | +/-0.01 | 0.945 | 0.96 | +/-0.06 | 0.369 |
| GBM | 0.97 | +/-0.01 | 1.445 | 0.93 | +/-0.11 | 0.439 |
| AdaBoost | 0.58 | +/-0.28 | 1.873 | 0.92 | +/-0.12 | 0.612 |
| LR | 0.85 | +/-0.01 | 0.598 | 0.88 | +/-0.07 | 0.084 |
| SVC | 0.86 | +/-0.01 | 11.423 | 0.91 | +/-0.11 | 10.922 |
| SVEC-H | 0.98 | +/-0.01 | 2.732 | 0.96 | +/-0.08 | 1.567 |
| SVEC-S | 0.98 | +/-0.01 | 3.181 | 0.96 | +/-0.08 | 0.972 |
| Proposed SSC | 0.98 | +/-0.01 | 3.012 | 0.99 | +/-0.01 | 3.705 |

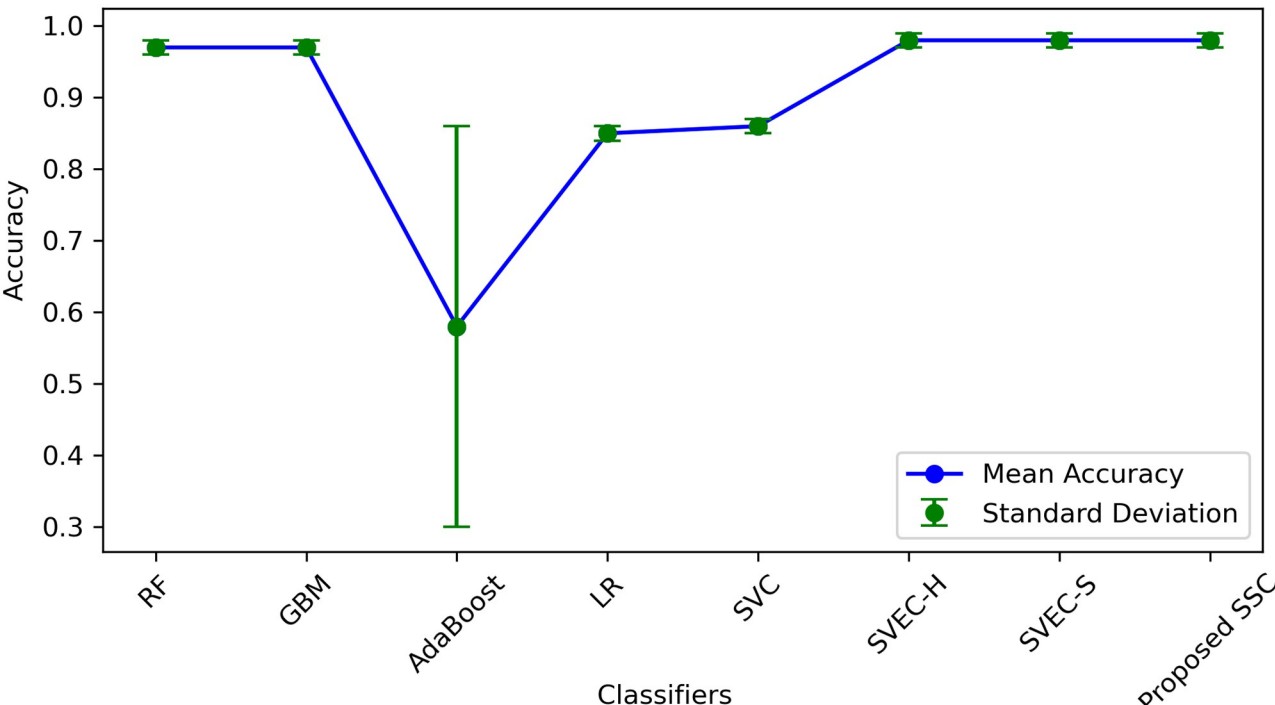

**Fig 11. Performance of models in terms of mean accuracy and error deviation.**

proposed classifier. The results of all machine learning classifiers with original and down-sampled data are presented in Table 13. The proposed SSC model outperformed other models with an accuracy of 0.99 and an SD of 0.01. Although SSC took longer to execute than some classifiers such as RF, GBM, AdaBoost, LR, and SVEC-S, the proposed model's accuracy has significantly improved. Therefore, despite the time difference, we can compromise it because the model generates highly accurate results, which is essential for medical diagnosis.

Figs 11 & 12, illustrates SD values for different classifiers and provides insights into the variability of accuracy scores when applied to both original and down-sampled data. It also shows the error rate lower and upper bounds. These values indicate the stability of classifier performance across multiple runs or datasets, with lower values suggesting more consistent results.

Finally, by comparing our results with those of other studies, we can see that, for the cited [35], the optimum results were attained by implementing machine learning-based feature selection and RF-based classification. However, in our research, balancing the dataset and employing SSC for classification contributed to the maximum performance. We also deployed the designed approaches cited in [19, 28, 48, 49] on our dataset to signify the performance of the proposed approach. Table 14 provides a detailed comparison of the results that illustrates the significance of the proposed SSC model. Given that we employed all features for the model training, SSC took longer to execute. Despite this, there is a significant difference in the cross-validation results of SSC and MLFS + RF proposed by Chaganti et al. [35]. This demonstrates better generalizability of the proposed re-sampling technique with SSC.

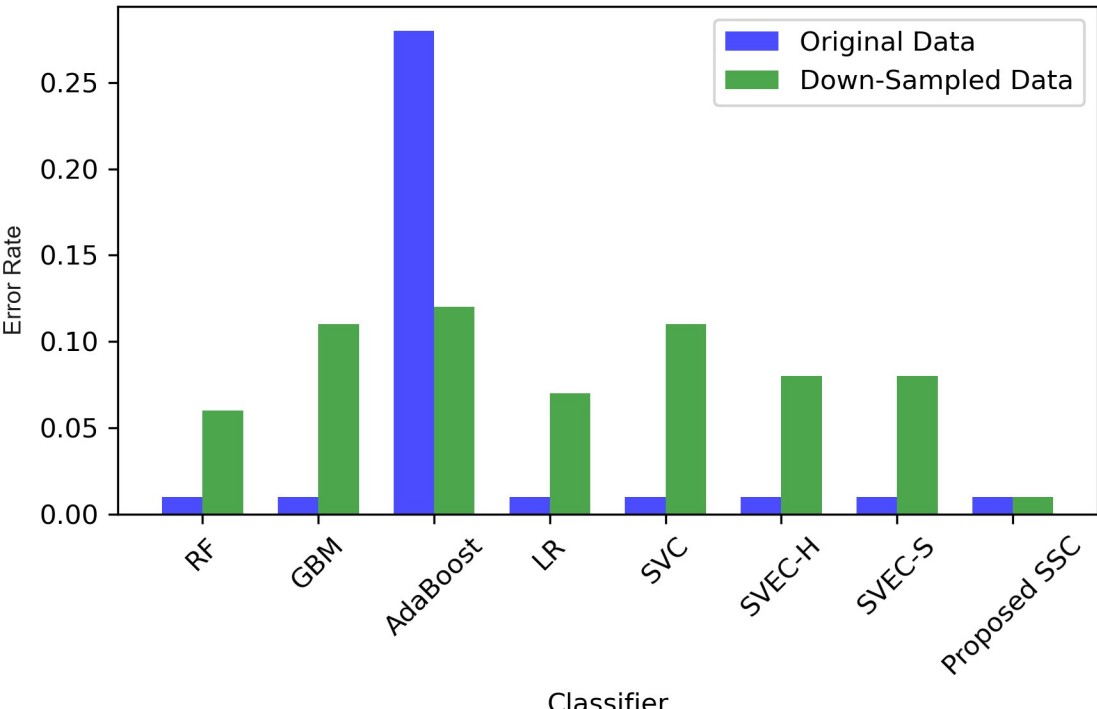

**Fig 12. Performance of models in terms of error rate.**

## 5.1 Statistical analysis

In this research, statistical analysis was performed using the T-test to compare the results of the proposed approach with other techniques. The T-test is a method to determine the statistical difference between two populations [50]. The aim was to investigate the statistical significance of the proposed approach in comparison to others. The null hypothesis is rejected by the T-test if the proposed approach is statistically significant. We set a p-value of 0.05 for all scenarios. The degree of freedom in each case is 7 and the results showed that the null hypothesis was rejected in all cases, as the P-value was less than the T scores. Therefore, the proposed approach was found to be statistically significant compared to others, as presented in Table 15.

To determine the statistical power of our test, we utilized an online power calculator [51]. Our sample size is 16, Cohen's effect size is 0.8, and the significance level is set at 0.05. The sample population consists of a normal distribution. The test results provide a statistical power

**Table 14. Performance comparison with previous study.**

| Study | Approach | Train-Test Method | | | | 10-Fold | |
|---|---|---|---|---|---|---|---|
| | | Accuracy | Precision | Recall | F1 | Accuracy | SD |
| [35] | MLFS + RF | 0.99 | 0.99 | 0.99 | 0.99 | 0.94 | ± 1.68 |
| [19] | RF | 0.98 | N/A | N/A | 0.98 | N/A | N/A |
| [49] | CNN + SGLV | 0.96 | N/A | N/A | 0.96 | N/A | N/A |
| [28] | DT | 0.98 | N/A | N/A | 0.97 | N/A | N/A |
| [48] | DT | 0.93 | N/A | N/A | 0.93 | N/A | N/A |
| Our | Down-sampling + SSC | 0.995 | 1.00 | 1.00 | 1.00 | 0.99 | ±0.01 |

**Table 15. Statistical T-test results.**

| Approach | T-Statistics |
|---|---|
| RF Vs SSC | 1.896 |
| GBM Vs SSC | 5.230 |
| ADA Vs SSC | 3.860 |
| LR Vs SSC | 3.666 |
| SVC Vs SSC | 2.759 |
| LSTM Vs SSC | 9.431 |
| CNN Vs SSC | 7.768 |
| CNN-LSTM Vs SSC | 8.038 |

of 0.169, with critical values of -1.96 and 1.96 based on the standard normal distribution, where the standard distribution shift is 2.263.

## 6 Conclusions

Medical expert systems for thyroid disease diagnosis have seen tremendous growth, and the systems currently available are sophisticated enough to be employed in practice, enhancing and improving patient care. The thyroid is particularly difficult to diagnose as its symptoms are often mistaken for those of other conditions. Early diagnosis of thyroid disease can prevent mishaps.

Machine learning models produce skewed results when provided with imbalanced datasets. Thyroid disease diagnosis requires an effective, accurate, and reliable system, and current diagnostic tools focus on reducing dataset dimensionality, ignoring the distribution of target classes. This study proposed a system that down-samples the dataset to obtain a balanced distribution of samples, classified using an RF-based self-stacking classifier (SSC). Experimental results show the proposed model's superior performance, achieving 99.5% accuracy, 100% macro precision, recall, and F1-score. The proposed classifier's performance was compared to several other machine learning, deep learning, and self-voting classifiers, demonstrating the reliability of the proposed approach through 10-fold cross-validation. The performance of SSC was compared to the existing medical diagnosis system, which corroborated the significance and superiority of this study. The following conclusions were drawn from the research:

- Balancing the dataset enables the classifiers to perform well with minority classes.

- Down-sampling substantially increases recall, resulting in a decline in the FP to TP ratio, indicating successful identification of minority data.

- For highly imbalanced datasets, RF can be used for best performance.

- RF-based self-stacking ensemble outperformed RF-based self-voting ensemble classifiers regarding the balanced dataset.

A limitation of this study is that each target class was down-sampled to 230 data instances, which we aim to address in ongoing research by experimenting with a more extensive distribution of data samples across each target variable.

## Author Contributions

**Conceptualization:** Shengjun Ji.

**Data curation:** Shengjun Ji.

**Formal analysis:** Shengjun Ji.

**Investigation:** Shengjun Ji.

**Methodology:** Shengjun Ji.

**Project administration:** Shengjun Ji.

**Resources:** Shengjun Ji.

**Software:** Shengjun Ji.

**Validation:** Shengjun Ji.

**Visualization:** Shengjun Ji.

**Writing – original draft:** Shengjun Ji.

**Writing – review & editing:** Shengjun Ji.

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
