## [Decision Letter · Decision Letter 0]

8 Feb 2023

PONE-D-22-32955SSC: The Novel Self-Stack Ensemble Model for Thyroid Disease PredictionPLOS ONE

Dear Dr. Ji,

Thank you for submitting your manuscript to PLOS ONE. After careful consideration, we feel that it has merit but does not fully meet PLOS ONE’s publication criteria as it currently stands. Therefore, we invite you to submit a revised version of the manuscript that addresses the points raised during the review process. Please submit your revised manuscript by Mar 25 2023 11:59PM. If you will need more time than this to complete your revisions, please reply to this message or contact the journal office at plosone@plos.org. Please include the following items when submitting your revised manuscript:A rebuttal letter that responds to each point raised by the academic editor and reviewer(s). You should upload this letter as a separate file labeled 'Response to Reviewers'.A marked-up copy of your manuscript that highlights changes made to the original version. You should upload this as a separate file labeled 'Revised Manuscript with Track Changes'.An unmarked version of your revised paper without tracked changes. You should upload this as a separate file labeled 'Manuscript'.If applicable, we recommend that you deposit your laboratory protocols in protocols.io to enhance the reproducibility of your results. Protocols.io assigns your protocol its own identifier (DOI) so that it can be cited independently in the future. For instructions see: https://journals.plos.org/plosone/s/submission-guidelines#loc-laboratory-protocols. Additionally, PLOS ONE offers an option for publishing peer-reviewed Lab Protocol articles, which describe protocols hosted on protocols.io. Read more information on sharing protocols at https://plos.org/protocols?utm_medium=editorial-email&utm_source=authorletters&utm_campaign=protocols.

We look forward to receiving your revised manuscript.

Kind regards,

Maciej Huk, Ph.D.

Academic Editor

PLOS ONE

Journal Requirements:

    "No"

   "NO authors have competing interests"

6. Please ensure that you refer to Figure 3 and 9 in your text as, if accepted, production will need this reference to link the reader to the figure.

7. We note you have included a table to which you do not refer in the text of your manuscript. Please ensure that you refer to Table 11 in your text; if accepted, production will need this reference to link the reader to the Table.

Additional Editor Comments:In particular three minor corrections are needed:It should be cleared why in figure 9, the sum of the numbers of each confusion matrix is different. For example, the sums of the first lines in the subgraphs (a)(b)(c) is 51, while the first lines of (e)(f) and (j) add up to only 35. This should not occur if the same test set was used in each case.It should be discussed if RF, SVC and LR models were trained with optimal values of parameters.Algorithms should be presented with proper indentation.

Reviewers' comments:

Reviewer's Responses to Questions

**Comments to the Author**

1. Is the manuscript technically sound, and do the data support the conclusions?

Reviewer #1: Yes

Reviewer #2: Yes

2. Has the statistical analysis been performed appropriately and rigorously? 

Reviewer #1: Yes

Reviewer #2: Yes

3. Have the authors made all data underlying the findings in their manuscript fully available?

Reviewer #1: Yes

Reviewer #2: Yes

4. Is the manuscript presented in an intelligible fashion and written in standard English?

Reviewer #1: Yes

Reviewer #2: Yes

5. Review Comments to the Author

Reviewer #1: The author uses the SSC algorithm to train and test on the open data set, and compares it with a variety of machine learning models. Please check the following questions:

1. In figure 9, why the sum of the numbers of each confusion matrix is different, for example, the sum of the first line of RF in the subgraph (a) is 51, while the first line of (j) CNN-LSTM adds up to only 35. This problem should not occur in the same test suite.

2. When training models such as 2.RF, SVC and LR, are the optimal parameters selected? if not, what are the considerations? Similarly, what hyper-parameters CNN and LSTM choose and what models they choose need to be supplemented and clarified.

Reviewer #2: Machine learning algorithms cannot properly identify and recognize minority data, and thus disease, because they were trained on unbalanced thyroid disease data. One approach to solving this issue is resampling, which adjusts the ratio between the target classes to fairly balance the data.

This paper proposes a novel RF-based self-stacking classifier for effective thyroid illness diagnosis is also presented. They use a downsampling strategy to attain a balanced distribution of target classes.

Their study uses the UCI thyroid illness dataset, which has 9172 samples, 30 characteristics, and a very unbalanced target class distribution.

They achieve 99.5% accuracy, and their suggested method may identify primary hypothyroidism, elevated binding protein, compensated hypothyroidism, and concurrent non-thyroidal disease. The proposed model produced 100% macro precision and 100% macro recall, demonstrating cutting-edge performance. The suggested model demonstrated state-of-the-art performance by producing 100% macro precision, 100% macro recall, and 100% macro f1-score. A full comparative analysis is carried out to show the practicality of the suggested approach. Numerous deep neural networks, ensemble voting classifiers, and machine learning classifiers were used to achieve this. K-fold cross-validation results demonstrate the effectiveness of the suggested self-stacking classifier.

Overall this is an interesting paper: well-written and well-motivated to an important problem of applied ML. The author compares their methods with several other standard ML methods. The authors should indent the presentation of the Algorithms properly and add the descriptions of S and C in the caption.

6. PLOS authors have the option to publish the peer review history of their article (what does this mean?). If published, this will include your full peer review and any attached files.

Reviewer #1: No

Reviewer #2: No

---

## [Author Response · Author response to Decision Letter 0]

16 Feb 2023

Editor Comments:

• Concern 1: It should be cleared why in figure 9, the sum of the numbers of each confusion matrix is different. For example, the sums of the first lines in the subgraphs (a)(b)(c) is 51, while the first lines of (e)(f) and (j) add up to only 35. This should not occur if the same test set was used in each case.

Response: We would like to thank the editor for the valuable comments to improve the quality of the manuscript. You rise a good point and help us to mention it in the manuscript for clarity. We have done separate data splitting (Training and Testing sets) for machine learning deep learning models. When splitting the dataset, we used the shuffle parameter which changes the number of instances for each target class but in both cases, the total numbers of test samples are the same which is 230. We did splitting separately for deep learning models because you can see the last layer of deep learning have 5 neurons so we have to feed target data after converting it into five variables so we did splitting again after the conversion of data for deep learning models and shuffle parameters change the count of for each category in the test dataset.

• Concern 2: It should be discussed if RF, SVC and LR models were trained with optimal values of parameters.

Response: We would like to thank the reviewer for the valuable comments to improve the quality of the manuscript. We select the best hyperparameter setting by tuning the models on specific value ranges. We added the tuning range of the model's hyperparameters value to discuss the selection of optimal values of parameters for models.

• Concern 3: Algorithms should be presented with proper indentation.

Response: We make changes in the manuscript according to the reviewer's suggestion. We make corrections in algorithms indentation.

Reviewer #1: 

Concern 1: In figure 9, why the sum of the numbers of each confusion matrix is different, for example, the sum of the first line of RF in the subgraph (a) is 51, while the first line of (j) CNN-LSTM adds up to only 35. This problem should not occur in the same test suite.

Response: We would like to thank the reviewer for the valuable comments to improve the quality of the manuscript. You rise a good point and help us to mention it in the manuscript for clarity. We have done separate splitting for machine learning deep learning models. When splitting the dataset we used the shuffle parameter which changes the number of instances for target classes but in both cases, the total numbers of test samples are the same which is 230. We did splitting separately for deep learning models because you can see the last layer of deep learning have 5 neurons so we have to feed target data after converting it into five variables so we did splitting again after the conversion of data for deep learning models and shuffle parameters change the count of for each category in the test dataset.

Concern 2: When training models such as 2.RF, SVC and LR, are the optimal parameters selected? if not, what are the considerations? Similarly, what hyper-parameters CNN and LSTM choose and what models they choose need to be supplemented and clarified.

Response: We would like to thank the reviewer for the valuable comments to improve the quality of the manuscript. We select the best hyperparameter setting by tuning the models on specific value ranges. We added the tuning range of the model's hyperparameters value to discuss the selection of optimal values of parameters for models.

While for LSTM and CNN models architecture and hyperparameters setting we added in Table 6. This hyperparameter setting is according to the literature. We study the researchers who worked on the same kinds of datasets and we used the same kind of state-of-the-art architectures to achieve significant accuracy.

Reviewer #2: 

Concern 1: Overall this is an interesting paper: well-written and well-motivated to an important problem of applied ML. The author compares their methods with several other standard ML methods. The authors should indent the presentation of the Algorithms properly and add the descriptions of S and C in the caption.

Response: We would like to thank the reviewer for the valuable comments to improve the quality of the manuscript. We add the proper indent in the presentation of the Algorithms and also describe the variables in the algorithms.

---

## [Decision Letter · Decision Letter 1]

11 Apr 2023

PONE-D-22-32955R1SSC: The Novel Self-Stack Ensemble Model for Thyroid Disease PredictionPLOS ONE

Dear Dr. Ji,

Thank you for submitting your manuscript to PLOS ONE. It was analyzed by three Reviewers including me as an Academic Editor (Reviewer #4). After careful consideration, we feel that it has merit but does not fully meet PLOS ONE’s publication criteria as it currently stands. Therefore, we invite you to submit a revised version of the manuscript that addresses the points raised during the review process. In particular:statistical analysis of results should be presented,language and presentation problems should be removed,experimental setup should be better justified or modified.

We look forward to receiving your revised manuscript.

Kind regards,

Maciej Huk, Ph.D.

Academic Editor

PLOS ONE

Reviewers' comments:

Reviewer's Responses to Questions

**Comments to the Author**

1. If the authors have adequately addressed your comments raised in a previous round of review and you feel that this manuscript is now acceptable for publication, you may indicate that here to bypass the “Comments to the Author” section, enter your conflict of interest statement in the “Confidential to Editor” section, and submit your "Accept" recommendation.

Reviewer #1: All comments have been addressed

Reviewer #3: (No Response)

Reviewer #4: (No Response)

2. Is the manuscript technically sound, and do the data support the conclusions?

Reviewer #1: Yes

Reviewer #3: Yes

Reviewer #4: Partly

3. Has the statistical analysis been performed appropriately and rigorously? 

Reviewer #1: Yes

Reviewer #3: Yes

Reviewer #4: No

4. Have the authors made all data underlying the findings in their manuscript fully available?

Reviewer #1: Yes

Reviewer #3: Yes

Reviewer #4: Yes

5. Is the manuscript presented in an intelligible fashion and written in standard English?

Reviewer #1: Yes

Reviewer #3: Yes

Reviewer #4: Yes

6. Review Comments to the Author

Reviewer #1: The performance of these models looks good, but an important point, how can it be explained? That is, the order of importance of each feature, such as shap. I strongly suggest adding, thank you.

Reviewer #3: The proposed Self Stacking classifier must be clarified with a figure. Also, there are others authors whose have used the same approach https://link.springer.com/article/10.1007/s12065-023-00824-4 using breast cancer. I think the model is not novel but the thyroid disease prediction model application is. Thus, the authors should redefine the manuscript accordingly with these ideas.

Reviewer #4: >>> 1. Language problems:

1.1 "i.e RF" => "i.e. RF" (x2)

1.2 "f1-score" => "F1-score"

1.3 "Where TP" => "where TP " (after eq. (4))

1.4 "Competing Intereset" => "Competing Interests"

1.5 "at the iot edge" => "at the IoT edge" (ref. 42)

>>> 2. Presentation problems:

2.1 Fig 1. Within stacked ensemble model - Model1 includes Model1?

2.2 Fig 1. "Trained Model" block is connected with "Model" block in "Test set" part.

The meaning of this connection is unclear. Is stacked ensemble model influencing the "Model"?

Maybe the "Trained Model" block should be connected to "Model Evaluation" block?

2.3 Table 3. Title is too general. Please at least specify dataset.

2.4 While Table 4 is presented then are Authors sure that Fig. 2 is needed or useful?

Both present the same very simple relation and fact.

2.5 Table 11. Title is too general. Please specify what neural networks are characterized within the table. Maybe Authors want to write "Classification results of example considered deep neural networks"

2.6 Fig. 2.5 and Fig. 2.6: Title includes a lot of repeated fragments what makes it very long - please reorganize to remove repeated fragments.

2.7 Table 13: units are not presented in the header of the table (time in seconds?)

2.8 Error bars on charts are not shown on charts. Table 13: time is given without confidence intervals (or standard dev.).

>>> 3. Other problems:

3.1 Why the plain model used for comparison is built with only 20% of data? This is suggested by Fig. 1 and seems to be unfair - ensemble and non-ensemble models before comparison should be built in the same experimental environment.

3.2 Fig. 8 (h), (i), (j) - rows of confusion matrices are ordered in different way than in the rest of models. This should be clarified and organized in unified way (the same classes should have the same row number designations in case of all models/matrices.

3.3 Sums of the same rows (the same classes) are not equal between matrices in Fig 7. The same problem with Fig 6.

3.4 It is hard to understand the meaning of "10-fold" column in Table 14. In fact, all considered measures should be calculated with use of the k-fold cross validation.

3.5 Statistical analysis of results is not presented.

3.6 It would be good to make software prepared for experimentation fully available for the Readers.

>>> Summary:

Manuscript includes a lot of presentation and other problems. Especially serious are those with experimental setup design (remark 3.1) and statistical analysis of presented results (remarks 3.4 and 3.5). Those errors should be fixed before publication.

Recommendation: major rework

===EOT===

7. PLOS authors have the option to publish the peer review history of their article (what does this mean?). If published, this will include your full peer review and any attached files.

Reviewer #1: No

Reviewer #3: **Yes: **Marlon Santiago Viñán Ludeña

Reviewer #4: No

---

## [Author Response · Author response to Decision Letter 1]

18 Apr 2023

Editor Comments: 

Thank you for submitting your manuscript to PLOS ONE. It was analyzed by three Reviewers including me as an Academic Editor (Reviewer #4). After careful consideration, we feel that it has merit but does not fully meet PLOS ONE’s publication criteria as it currently stands. Therefore, we invite you to submit a revised version of the manuscript that addresses the points raised during the review process.

In particular:

• statistical analysis of results should be presented,

• language and presentation problems should be removed,

• experimental setup should be better justified or modified.

Response: We would like to thank the reviewer for giving us a chance to work on the weakness in the manuscript. We work on all comments such as we added the statistical analysis, and we have done work on language and presentation of data in the manuscript. We also explain the experimental setup in more detail in the updated manuscript.

Statistical Analysis: In this research, statistical analysis was performed using the T-test to compare the results of the proposed approach with other techniques. The T-test is a method to determine the statistical difference between two populations [51]. The aim was to investigate the statistical significance of the proposed approach in comparison to others. The null hypothesis is rejected by the T-test if the proposed approach is statistically significant. The rejection is based on the P-value and alpha value, which in this study was set at 0.5 for all scenarios. The results showed that the null hypothesis was rejected in all cases, as the P-value was less than the alpha value. Therefore, the proposed approach was found to be statistically significant compared to others, as presented in Table 15.

The remaining changes related to language and presentation issues can be found in the updated manuscript.

Reviewer #1: 

Concern 1: The performance of these models looks good, but an important point, how can it be explained? That is, the order of importance of each feature, such as shap. I strongly suggest adding, thank you.

Response: We would like to thank the reviewer for the valuable comment to improve the quality of the manuscript. We added the feature importance score in the paper which show why the machine learning model's performance is good after downsampling.

In Figure 2, the feature importance scores are displayed for both the original dataset and the down-sampled dataset. These scores were obtained using the Extra Trees Classifier (ETC), which takes all features and the target into account for both scenarios. The ETC is a tree-based classifier that determines the importance of each feature by calculating its entropy criterion. The feature importance scores for both scenarios are highlighted in Figures 2a and 2b. It is observed that there is only a minor difference between the feature importance scores for both scenarios and the range of feature importance scores has increased after balancing the data which helps to improve the performance of learning models in this study.

Reviewer #3: 

Concern 1: The proposed Self Stacking classifier must be clarified with a figure. Also, there are others authors whose have used the same approach https://link.springer.com/article/10.1007/s12065-023-00824-4 using breast cancer. I think the model is not novel but the thyroid disease prediction model application is. Thus, the authors should redefine the manuscript accordingly with these ideas.

Response: We would like to thank the reviewer for raising this concern and giving us a chance to elaborate on the difference. The mentioned study worked on the staked classifier but there are all based learners are different models while we give it the name self-learner because all base models are RF and even the meta-model is also the same model RF which means RF is learning from itself through base models which make it self learner. While the mentioned study called used the SELF keyword because they used staked ensemble learning framework (SELF). So there is clear a difference between our approach and their approach.

Reviewer #4: 

Concern 1: >>> 1. Language problems:

1.1 "i.e RF" => "i.e. RF" (x2)

1.2 "f1-score" => "F1-score"

1.3 "Where TP" => "where TP " (after eq. (4))

1.4 "Competing Intereset" => "Competing Interests"

1.5 "at the iot edge" => "at the IoT edge" (ref. 42)

Response: We would like to thank the reviewer for highlighting the issues in the manuscript, we update the manuscript by removing all typos and mistakes.

Concern 2: >>> 2. Presentation problems:

2.1 Fig 1. Within stacked ensemble model - Model1 includes Model1?

2.2 Fig 1. "Trained Model" block is connected with "Model" block in "Test set" part.

The meaning of this connection is unclear. Is stacked ensemble model influencing the "Model"?

Maybe the "Trained Model" block should be connected to "Model Evaluation" block?

Response: We make changes according to reviewer concerns. Model1 replace with word meta models and the evaluation block is attached to the trained model.

2.3 Table 3. Title is too general. Please at least specify dataset.

Response: We update the caption for Table 3 according to the reviewer suggestion.

2.4 While Table 4 is presented then are Authors sure that Fig. 2 is needed or useful?

Both present the same very simple relation and fact.

Response: Reviewer's suggestion is valuable and we remove Figure 2 from the paper to avoid data redundancies in the manuscript. 

2.5 Table 11. Title is too general. Please specify what neural networks are characterized within the table. Maybe Authors want to write "Classification results of example considered deep neural networks"

Response: We update the caption for Table 11 according to the reviewer suggestion.

2.6 Fig. 2.5 and Fig. 2.6: Title includes a lot of repeated fragments what makes it very long - please reorganize to remove repeated fragments.

Response: We update the caption for Figures according to the reviewer suggestion.

2.7 Table 13: units are not presented in the header of the table (time in seconds?)

Response: We measure the computation cost of machine learning models in terms of time (second). We update the manuscript according to the reviewer suggestion.

2.8 Error bars on charts are not shown on charts. Table 13: time is given without confidence intervals (or standard dev.).

Response: We update the manuscript according to the reviewer suggestion.

Concern 3: >>> 3. Other problems:

3.1 Why the plain model used for comparison is built with only 20% of data? This is suggested by Fig. 1 and seems to be unfair - ensemble and non-ensemble models before comparison should be built in the same experimental environment.

Response: We would like to thank the reviewer for the valuable comments. We deploy all models in the same environment and give the same size of data to all models. We update the Fig 1 to avoid confusion. All models are trained on 80% training data and then evaluation is done using 20% test data.

3.2 Fig. 8 (h), (i), (j) - rows of confusion matrices are ordered in different way than in the rest of models. This should be clarified and organized in unified way (the same classes should have the same row number designations in case of all models/matrices.

Response: We would like to thank the reviewer for the valueable comments. The order is the same for all confusion matrices. The difference which you can see is because of separate data splitting for machine learning and deep learning models. When splitting the dataset we used the shuffle parameter which changes the number of instances for target classes but in both cases, the total numbers of test samples are the same which is 230. We did splitting separately for deep learning models because you can see the last layer of deep learning have 5 neurons so we have to feed target data after converting it into five variables we did splitting again after the conversion of data for deep learning models and shuffled parameters change the count of for each category in the test dataset.

3.3 Sums of the same rows (the same classes) are not equal between matrices in Fig 7. The same problem with Fig 6.

Response: We would like to thank the reviewer for the valuable comments to improve the quality of the manuscript. You rise a good point and help us to mention it in the manuscript for clarity. We have done separate data splitting (Training and Testing sets) for machine learning deep learning models. When splitting the dataset, we used the shuffle parameter which changes the number of instances for each target class but in both cases, the total numbers of test samples are the same which is 230. We did splitting separately for deep learning models because you can see the last layer of deep learning have 5 neurons so we have to feed target data after converting it into five variables we did splitting again after the conversion of data for deep learning models and shuffled parameters change the count of for each category in the test dataset.

3.4 It is hard to understand the meaning of "10-fold" column in Table 14. In fact, all considered measures should be calculated with use of the k-fold cross validation.

Response: Table 14 contain accuracy, precision, recall, and F1 score which we find using test data, while 10-fold accuracy and SD using the 10-fold cross-validation approach. We update the table structure for clear understanding.

3.5 Statistical analysis of results is not presented.

Response: We would like to thank the reviewer for the valuable comment to improve the quality of the manuscript. We added the statistical analysis in the updated manuscript.

In this research, statistical analysis was performed using the T-test to compare the results of the proposed approach with other techniques. The T-test is a method to determine the statistical difference between two populations [51]. The aim was to investigate the statistical significance of the proposed approach in comparison to others. The null hypothesis is rejected by the T-test if the proposed approach is statistically significant. The rejection is based on the P-value and alpha value, which in this study was set at 0.5 for all scenarios. The results showed that the null hypothesis was rejected in all cases, as the P-value was less than the alpha value. Therefore, the proposed approach was found to be statistically significant compared to others, as presented in Table 15.

3.6 It would be good to make software prepared for experimentation fully available for the Readers.

Response: We would like to thank the reviewer for the suggestion. We will make the experiment's code public and will add a link to the code repository in the paper after the paper's acceptance. 

Concern 4: Manuscript includes a lot of presentation and other problems. Especially serious are those with experimental setup design (remark 3.1) and statistical analysis of presented results (remarks 3.4 and 3.5). Those errors should be fixed before publication.

Response: Reviewer suggestions to improve the manuscript are highly appreciated. We resolve are comments which mention above such as adding experimental details more clearly, adding statistical Test results, and checking Grammarly thoroughly.

---

## [Decision Letter · Decision Letter 2]

29 May 2023

PONE-D-22-32955R2SSC: The Novel Self-Stack Ensemble Model for Thyroid Disease PredictionPLOS ONE

Dear Dr. Ji,

Thank you for submitting your manuscript to PLOS ONE. It war reviewed by three Reviewers including me as an Academic Editor (Reviewer #4). After careful consideration, we feel that it has merit but does not fully meet PLOS ONE’s publication criteria as it currently stands. Therefore, we invite you to submit a revised version of the manuscript that addresses the points raised during the review process. In particular:statistical analysis and reporting of results should be improved,presentation of results should be fixed (including results and legends in figures),measurement setup should be described clearly (including measurements of time),pseudocode of algorithm 2 should be made clear.Please submit your revised manuscript by Jul 13 2023 11:59PM. If you will need more time than this to complete your revisions, please reply to this message or contact the journal office at plosone@plos.org. Please include the following items when submitting your revised manuscript:A rebuttal letter that responds to each point raised by the academic editor and reviewer(s). You should upload this letter as a separate file labeled 'Response to Reviewers'.A marked-up copy of your manuscript that highlights changes made to the original version. You should upload this as a separate file labeled 'Revised Manuscript with Track Changes'.An unmarked version of your revised paper without tracked changes. You should upload this as a separate file labeled 'Manuscript'.

We look forward to receiving your revised manuscript.

Kind regards,

Maciej Huk, Ph.D.

Academic Editor

PLOS ONE

Reviewers' comments:

Reviewer's Responses to Questions

**Comments to the Author**

1. If the authors have adequately addressed your comments raised in a previous round of review and you feel that this manuscript is now acceptable for publication, you may indicate that here to bypass the “Comments to the Author” section, enter your conflict of interest statement in the “Confidential to Editor” section, and submit your "Accept" recommendation.

Reviewer #1: All comments have been addressed

Reviewer #3: All comments have been addressed

Reviewer #4: (No Response)

2. Is the manuscript technically sound, and do the data support the conclusions?

Reviewer #1: Yes

Reviewer #3: Yes

Reviewer #4: Partly

3. Has the statistical analysis been performed appropriately and rigorously? 

Reviewer #1: Yes

Reviewer #3: Yes

Reviewer #4: No

4. Have the authors made all data underlying the findings in their manuscript fully available?

Reviewer #1: Yes

Reviewer #3: Yes

Reviewer #4: Yes

5. Is the manuscript presented in an intelligible fashion and written in standard English?

Reviewer #1: Yes

Reviewer #3: Yes

Reviewer #4: Yes

6. Review Comments to the Author

Reviewer #1: The author replied to my question and gave a better explanation. Based on the revised manuscript, I think it meets the relevant requirements for publication.

Reviewer #3: Specify what acronyms such as RF stand for. Lectors must knowing what it means when acronym appear at the first time

Reviewer #4: >>> 1. Language problems:

1.1 "Classifier_base = Base Classifiers i.e RF_base1, RF_base2, and RF_base3" (in Algorithm 2)

"i.e" => "i.e."

>>> 2. Presentation problems:

2.1 Table 11, Table 13, Table 14: units are not presented in the header of the table (accuracy in [%] or [1]). SD for accuracy and other measured values is not given.

2.2 Fig. 2a and Fig. 2b: most of the names of features near horizontal axis are presented improperly. Those names are cut in a way that makes some of those features indistinguishable (e.g. 6x"sured", 2x"oxine", 2x"yroid". This should not happen.

2.3 Fig. 9, Fig. 5, Fig 6, Fig 2: Results should be presented with error bars.

2.4 Table 15: There is no point presenting columns with the same value in all rows of the table. Such information can be given in the title of the table or below the table.

2.5 Algorithm 2:

"D = Number of Classifier_base used." - it is not clear

=>

"D = Number of base classifiers."

2.6 Fig. 5: what is the meaning of "Val_Accuracy", "Val_Precision", "Val_Recall" and 'Val_F1" within legends of included charts? What is the difference e.g. between "Accuracy' and ""Val_Accuracy"? This should be explained within the title of the figure.

2.7 Fig. 6 - analogous problem as the one indicated in point 2.6. What is e.g. Loss and how it is different from Val_Loss?

>>> 3. Other problems:

3.1 Why the time in Table 13 is given with varying precision for different models (please compare results for SVC v.s. GBM v.s. LR)? Were different methods for time measurement used in those cases? It should be specified within the text how the time was measured and what was the precision of this time measurement setup. SD or confdence intervals should be given with measured value.

3.2 Algorithm 2: It is unclear what it the meaning of "mode{}". Maybe Authors meant "model" ?

In both cases it should be clarified how the prediction is generated in the last step of Algorithm 2.

3.3 Fig. 8 (h), (i), (j) - rows of confusion matrices are ordered in different way than in the rest of models. This should be clarified and organized in unified way (the same classes should have the same row number designations in case of all models/matrices. E.g. Fig. 8 (h), (i), (j) row 0 include 35 hits while other confusion matrices in Fig 8 include 35 hits in row number 3. Such changes in class identifiers make this figure hard to understand.

This problem should be removed.

3.4 Sums of the same rows (the same classes) are not equal between matrices in Fig 7.

If this is not an error then it should be explained in more clear way.

3.5 Authors write: "All experiments were performed using a Corei7, a 12th generation Dell machine with a Windows operating system."

Are the names of the hardware producer, processor and operating system important for analysis of the results? If yes, this should be clearly discussed within teh text. If not, then not needed information should be removed.

3.6 Authors write: "The rejection is based on the P-value and alpha value, which in this study was set at 0.5 for all scenarios."

Alpha value equal 0.5 represents very weak test. Maybe Authors meant 0.05?

It should be given what was the number of degrees of freedom during calculation of t-statistics? 229?

Why the p-value in case of all comparisons is the same? Is this an effect of imprecision of IEEE754 format used during calculations?

>>> Recommendation: major rework

===EOT===

7. PLOS authors have the option to publish the peer review history of their article (what does this mean?). If published, this will include your full peer review and any attached files.

Reviewer #1: No

Reviewer #3: No

Reviewer #4: No

---

## [Author Response · Author response to Decision Letter 2]

26 Jun 2023

RESPONSE TO REVIEWERS

Manuscript ID: PONE-D-22-32955

Article Title: “The Novel Self-Stack Ensemble Model for Thyroid Disease Prediction” 

Author: Ji Sheng-jun

Dear Editor, 

First, we are extremely grateful to the editor and all the reviewers who conducted a rigorous and constructive review of our article number PONE-D-22-32955. Certainly, all the review points were very well directed, and we managed to improve our manuscript a great deal and had to work a lot to address all the changes suggested by the reviewers. We have incorporated suggestions and guidance provided by the reviewers and made a careful revision of the manuscript. We have tried our level best to improve this manuscript by organizing and improving our figures to present the material more constructively and effectively. This has certainly, raised the quality of this research work and will make it a valuable contribution to the community. We hope that we have managed to fulfil the required changes suggested by the valuable reviewers. 

Please see our responses (in blue color) in line with the reviewers’ comments in black color. 

Best regards,

Ji Sheng-jun

 

Reviewer #1: 

Concern: The author replied to my question and gave a better explanation. Based on the revised manuscript, I think it meets the relevant requirements for publication.

Response: We would like to thank the reviewer for value comments to improve the quality of manuscript and giving positive feedback.

Reviewer #3: 

Concern: Specify what acronyms such as RF stand for. Lectors must knowing what it means when acronym appear at the first time.

Response: We would like to thank the reviewer for highlighting an important point and we update manuscript according to reviewer concern. We added the acronyms for all abbreviation. 

Reviewer #4: 

Concern 1: >>> 1. Language problems:

1.1 "Classifier_base = Base Classifiers i.e RF_base1, RF_base2, and RF_base3" (in Algorithm 2)

"i.e" => "i.e."

We would like to thank the reviewer for highlighting an important point and we update manuscript according to reviewer concern.

Concern 2: >>> 2. Presentation problems:

2.1 Table 11, Table 13, Table 14: units are not presented in the header of the table (accuracy in [%] or [1]). SD for accuracy and other measured values is not given.

Response: We would like to thank the reviewer for valuable comments. In Table 11, 13, and 14 and all others Tables we preset results in standard format such as accuracy and other evaluation are between 0 and 1. Our results in Tables are not in %. Second the standard deviation (SD) is output of K-fold cross validation and is present in corresponding Table 13.

Concern 3: 2.2 Fig. 2a and Fig. 2b: most of the names of features near horizontal axis are presented improperly. Those names are cut in a way that makes some of those features indistinguishable (e.g. 6x"sured", 2x"oxine", 2x"yroid". This should not happen.

Response: We would like to thank the reviewer for highlighting an important point and we update manuscript according to reviewer concern. We update the manuscript by adding the accurate images. 

2.3 Fig. 9, Fig. 5, Fig 6, Fig 2: Results should be presented with error bars.

Response: We would like to thank the reviewer for valuable comments.

• Fig 2: its feature importance figure so their cant be any error bar.

• Fig 5 and Fig 6: These two figures consist of accuracy and error graphs, in Fig 5 we illustrate the accuracies graph while in Fig 6 there are error graphs.

• Fig 7 & 8: Both figures consist of confusion matrix which values represent the error rates and accuracies both.

• Fig 9: we added error rate in fig 9 and update the manuscript.

2.4 Table 15: There is no point presenting columns with the same value in all rows of the table. Such information can be given in the title of the table or below the table.

Response: We would like to thank the reviewer for valuable comments to improve the quality of manuscript. The P-value column value can be different in each row but in our case it constant. We have T-statistics with P-values to evaluate the performance. So according to situate its better to have value in each row but still if reviewer will suggest we will remove this.

2.5 Algorithm 2:

"D = Number of Classifier_base used." - it is not clear

=>

"D = Number of base classifiers."

Response: We would like to thank the reviewer for valuable comments to improve the quality of manuscript. We update manuscript according to reviewer suggestion.

2.6 Fig. 5: what is the meaning of "Val_Accuracy", "Val_Precision", "Val_Recall" and 'Val_F1" within legends of included charts? What is the difference e.g. between "Accuracy' and ""Val_Accuracy"? This should be explained within the title of the figure.

Response: We would like to thank the reviewer for the valuable comments to improve the quality of the manuscript. The ‘Accuracy’ represent the accuracy on the training set and ‘Val_Accuracy’ represent the accuracy on the validation set, it is also similar to other evaluation matrixes. We added the description of it in the updated manuscript.

The meanings of 'Accuracy' and 'Val\\_Accuracy' in Figures 5 and 6 refer to the accuracy values obtained on the training and validation sets, respectively. The same principle applies to other evaluation measures as well.

2.7 Fig. 6 - analogous problem as the one indicated in point 2.6. What is e.g. Loss and how it is different from Val_Loss?

Response: Its very similar to as we mention in response of 2.6, The ‘Loss represent the accuracy on the training set and ‘Val_Loss’ represent the accuracy on the validation set, it is also similar to other evaluation matrixes.

Concern: 3 >>> 3. Other problems:

3.1 Why the time in Table 13 is given with varying precision for different models (please compare results for SVC v.s. GBM v.s. LR)? Were different methods for time measurement used in those cases? It should be specified within the text how the time was measured and what was the precision of this time measurement setup. SD or confdence intervals should be given with measured value.

Response: We would like to thank the reviewer for the valuable comments to improve the quality of the manuscript. These all used models are different in their approaches so their computation time can vary on different run. We added the specification of system used for experiments.

All experiments were performed using a Corei7, a 12th-generation Dell machine with a Windows operating system. We used Python language to implement the proposed approach in Jupyter Notebook. Several libraries such as sci-kit learn, TensorFlow, and Keras were used for experimental purposes. 

SD for each model is present in Table 13 which show the confidence interval of model.

3.2 Algorithm 2: It is unclear what it the meaning of "mode{}". Maybe Authors meant "model" ?

In both cases it should be clarified how the prediction is generated in the last step of Algorithm 2.

Response: We would like to thank the reviewer for the valuable comments to improve the quality of the manuscript. Mode in SVC-H algorithm, represent that the target class with highest vote will be the final prediction and argmax in SVC-S, represent that the class with highest average probability will be final prediction.

3.3 Fig. 8 (h), (i), (j) - rows of confusion matrices are ordered in different way than in the rest of models. This should be clarified and organized in unified way (the same classes should have the same row number designations in case of all models/matrices. E.g. Fig. 8 (h), (i), (j) row 0 include 35 hits while other confusion matrices in Fig 8 include 35 hits in row number 3. Such changes in class identifiers make this figure hard to understand.

This problem should be removed.

Response: We would like to thank the reviewer for the valuable comments. The order is the same for all confusion matrices. The difference which you can see is because of separate data splitting for machine learning and deep learning models. When splitting the dataset we used the shuffle parameter which changes the number of instances for target classes but in both cases, the total numbers of test samples are the same which is 230. We did splitting separately for deep learning models because you can see the last layer of deep learning have 5 neurons so we have to feed target data after converting it into five variables we did splitting again after the conversion of data for deep learning models and shuffled parameters change the count of for each category in the test dataset.

3.4 Sums of the same rows (the same classes) are not equal between matrices in Fig 7.

If this is not an error then it should be explained in more clear way.

Response: We would like to thank the reviewer for the valuable comments. We did the separate splitting for both machine learning and deep learning models, because of this there is a difference which we clearly mention in the manuscript and make proposed justification in the 3.2 response.

3.5 Authors write: "All experiments were performed using a Corei7, a 12th generation Dell machine with a Windows operating system."

Are the names of the hardware producer, processor and operating system important for analysis of the results? If yes, this should be clearly discussed within teh text. If not, then not needed information should be removed.

Response: We express our gratitude to the reviewer for their valuable feedback. We have included the details of our experimental setup. These details are crucial for the reproducibility of our results as we not only measured the model's performance in terms of accuracy but also in terms of time.

3.6 Authors write: "The rejection is based on the P-value and alpha value, which in this study was set at 0.5 for all scenarios."

Alpha value equal 0.5 represents very weak test. Maybe Authors meant 0.05?

It should be given what was the number of degrees of freedom during calculation of t-statistics? 229?

Why the p-value in case of all comparisons is the same? Is this an effect of imprecision of IEEE754 format used during calculations?

Response: We would like to thank the reviewer for the valuable comments. According to reviewer suggestion we did experiment with p value 0.05 and also add the degree of freedom for each comparison. Still with this 0.05 p value our proposed technique reject null hypothesis which show that our proposed approach is statistically significant in comparison with other approaches for disease diagnostics.

---

## [Decision Letter · Decision Letter 3]

4 Aug 2023

PONE-D-22-32955R3SSC: The Novel Self-Stack Ensemble Model for Thyroid Disease PredictionPLOS ONE

Dear Dr. Ji,

Thank you for submitting your manuscript to PLOS ONE. It was reviewed by the Academic Editor (Reviewer #4). After careful consideration, we feel that it has merit but does not fully meet PLOS ONE’s publication criteria as it currently stands. Therefore, we invite you to submit a revised version of the manuscript that addresses the points raised during the review process. In particular:reporting of results within tables should be improved,presentation of results within figures should be fixed (including results and legends),precision of performed time measurements should be presented.Please submit your revised manuscript by Sep 18 2023 11:59PM. If you will need more time than this to complete your revisions, please reply to this message or contact the journal office at plosone@plos.org. Please include the following items when submitting your revised manuscript:A rebuttal letter that responds to each point raised by the academic editor and reviewer(s). You should upload this letter as a separate file labeled 'Response to Reviewers'.A marked-up copy of your manuscript that highlights changes made to the original version. You should upload this as a separate file labeled 'Revised Manuscript with Track Changes'.An unmarked version of your revised paper without tracked changes. You should upload this as a separate file labeled 'Manuscript'.

We look forward to receiving your revised manuscript.

Kind regards,

Maciej Huk, Ph.D.

Academic Editor

PLOS ONE

Reviewers' comments:

Reviewer's Responses to Questions

**Comments to the Author**

1. If the authors have adequately addressed your comments raised in a previous round of review and you feel that this manuscript is now acceptable for publication, you may indicate that here to bypass the “Comments to the Author” section, enter your conflict of interest statement in the “Confidential to Editor” section, and submit your "Accept" recommendation.

Reviewer #4: (No Response)

2. Is the manuscript technically sound, and do the data support the conclusions?

Reviewer #4: Partly

3. Has the statistical analysis been performed appropriately and rigorously? 

Reviewer #4: I Don't Know

4. Have the authors made all data underlying the findings in their manuscript fully available?

Reviewer #4: Yes

5. Is the manuscript presented in an intelligible fashion and written in standard English?

Reviewer #4: Yes

6. Review Comments to the Author

Reviewer #4: >>> 1. Language problems: not detected

>>> 2. Presentation problems:

2.1 Table 11, Table 13, Table 14: units are not presented in the header of the table (accuracy in [%] or [1]).

This was indicated previously but not fixed. ACC=0.97 can mean e.g. 0.97% or 97%

2.2 Fig. 9, Fig. 5, Fig 6, Fig 2: Results should be presented with error bars.

Fig. 2, Fig. 5, Fig. 6: Not fixed.

Fig. 9 - modified but not fixed:

Classification Error (1 - Classification Accuracy) was added/presented by the Authors within Fig. 9 but this is not what was meant by the Reviewer. Measurement errors need to be presented instead, e.g. for p-value=0.95.

If Authors want to use bar chart then bar chart with error bars should be presented.

Please see the exmple discussions of the measurement error/measurement uncertainty and related confidence intervals:

https://blogs.sas.com/content/iml/2019/10/09/statistic-error-bars-mean.html

https://www.r-bloggers.com/2021/06/error-bar-plot-in-r-adding-error-bars-quick-guide/

2.3 Table 15: There is no point presenting columns with the same value in all rows of the table. Such information can be given in the title of the table or below the table.

Not fixed. After modifications Table 15 still includes three columns (df, P-Value, Null Hypothesis) which include the same values in each row.

2.4 Algorithm 2:

"D = Number of base classifier." => "D = Number of base classifiers."

Are Authors sure that plural version would not be more accurate?

2.5 Fig. 5, Fig. 6: what is the meaning of "Val_Accuracy", "Val_Precision", "Val_Recall", "Val_F1", "Val_Loss", "Val_MSE" within legends of included charts?

What is the difference e.g. between "Accuracy' and ""Val_Accuracy"? This should be explained within the title of the figure.

Authors write in their response:"‘Val_Accuracy’ represent the accuracy on the validation set, it is also similar to other

evaluation matrixes. We added the description of it in the updated manuscript."

The information was added within the text. It would be better to add relevant information within titles of Fig 5 and Fig 6.

The Reader can have a problem to find this information while analysing those figures.

>>> 3. Other problems:

3.1 Why the time in Table 13 is given with varying precision for different models?

What was the precision of time measurement setup? SD or confdence intervals should be given with measured value.

3.2 Fig. 8 (h), (i), (j) - rows of confusion matrices are ordered in different way than in the rest of models. This should be clarified and organized in unified way (the same classes should have the same row number designations in case of all models/matrices. E.g. Fig. 8 (h), (i), (j) row 0 include 35 hits while other confusion matrices in Fig 8 include 35 hits in row number 3. Such changes in class identifiers make this figure hard to understand.

This problem should be removed.

Authors gave the explanation within their response to the Reviewer.

The explanation should be added within the text of the manuscript it must be available to the Reader.

3.3 Sums of the same rows (the same classes) are not equal between matrices in Fig 7.

If this is not an error then it should be explained in more clear way.

Authors gave the explanation within their response to the Reviewer.

The explanation should be added within the text of the manuscript it must be available to the Reader.

>>> Recommendation: major rework

===EOT===

7. PLOS authors have the option to publish the peer review history of their article (what does this mean?). If published, this will include your full peer review and any attached files.

Reviewer #4: No

---

## [Author Response · Author response to Decision Letter 3]

9 Sep 2023

Reviewer #4:

Concern 1: >>> 2. Presentation problems: 2.1 Table 11, Table 13, Table 14: units are not presented in the header of the table (accuracy in [%] or [1]).

This was indicated previously but not fixed. ACC=0.97 can mean e.g. 0.97% or 97%

Response: We would like to thank the reviewer for raising concerns. Here, we added clear details about score units, as these accuracy, precision, recall, and F1 scores are standard parameters, so their values are always between 0 and 1. Here, 0 is the lowest score and 1 is the highest score. There will be no unit with them. 

We added this explanation in the manuscript for readers in the evaluation matrix section.

Concern 2: 2.2 Fig. 9, Fig. 5, Fig 6, Fig 2: Results should be presented with error bars.

Response: We would like to thank the reviewer for the valuable comments. We already responded to these comments in our previous response. Again we clarify this comment point by point. We also added one more figure (Figure 10) to illustrate error bars.

• Fig 2: This figure is already a bar chart but we can’t convert it into an error bar. Because of its feature importance score always one value corresponds to one feature. 

• Fig 5 and Fig 6: We convert these figures into bar charts as shown in Figure 9. These two figures consist of accuracy and error graphs so we convert them into bar charts and illustrate them in one graph.

• Fig 9: We added the error rate in Fig 9 and error bars in Fig 10 and updated the manuscript.

Concern 3: Fig. 2, Fig. 5, Fig. 6: Not fixed.

Response: We fixed these figures and converted them according to the reviewer concern. 

Concern 4: Fig. 9 - modified but not fixed: Classification Error (1 - Classification Accuracy) was added/presented by the Authors within Fig. 9 but this is not what was meant by the Reviewer. Measurement errors need to be presented instead, e.g. for p-value=0.95.

If Authors want to use bar chart then bar chart with error bars should be presented.

Please see the exmple discussions of the measurement error/measurement uncertainty and related confidence intervals:

https://blogs.sas.com/content/iml/2019/10/09/statistic-error-bars-mean.html

https://www.r-bloggers.com/2021/06/error-bar-plot-in-r-adding-error-bars-quick-guide/

Response: Figure 10, illustrates SD values for different classifiers and provides insights into the variability of accuracy scores when applied to both original and down-sampled data. It also shows the error rate lower and upper bounds. These values indicate the stability of classifier performance across multiple runs or datasets, with lower values suggesting more consistent results.

Concern 5: 2.3 Table 15: There is no point presenting columns with the same value in all rows of the table. Such information can be given in the title of the table or below the table.

Not fixed. After modifications Table 15 still includes three columns (df, P-Value, Null Hypothesis) which include the same values in each row.

Response: We remove the values and mention the values in the text.

Concern 6:2.4 Algorithm 2:

"D = Number of base classifier." => "D = Number of base classifiers."

Are Authors sure that plural version would not be more accurate?

Response: Yes, the number of base classifiers is correct and we update it accordingly.

Concern 7: 2.5 Fig. 5, Fig. 6: what is the meaning of "Val_Accuracy", "Val_Precision", "Val_Recall", "Val_F1", "Val_Loss", "Val_MSE" within legends of included charts?

What is the difference e.g. between "Accuracy' and ""Val_Accuracy"? This should be explained within the title of the figure.

Authors write in their response:"‘Val_Accuracy’ represent the accuracy on the validation set, it is also similar to other

evaluation matrixes. We added the description of it in the updated manuscript."

The information was added within the text. It would be better to add relevant information within titles of Fig 5 and Fig 6.

The Reader can have a problem to find this information while analysing those figures.

Response: We updated the manuscript according to the reviewer's suggestion and added details in the text and captions.

The terms 'Accuracy' and 'Val_Accuracy' in Figures 5 and 6 correspond to the accuracy metrics computed on the training and validation datasets, respectively. This same concept extends to other evaluation metrics, where 'val_precision' represents validation precision, 'val_recall' stands for validation recall, and 'val_F1_Score' stands for the validation F1 score.

Concern 8: 3.1 Why the time in Table 13 is given with varying precision for different models?

What was the precision of time measurement setup? SD or confdence intervals should be given with measured value.

Response: Reviewers' concern is valid, and we would like to address it. The variation in processing time can be attributed to differences in model architectures. For instance, linear models like Support Vector Classifier (SVC) demand more computational resources and time to find an optimal hyperplane, whereas models like Random Forest (RF), Logistic Regression (LR), and Gradient Boosting Machine (GBM) have less complex architectures compared to SVC. Therefore, the time precision is entirely dependent on the specific model's architecture.

Concern 9: 3.2 Fig. 8 (h), (i), (j) - rows of confusion matrices are ordered in different way than in the rest of models. This should be clarified and organized in unified way (the same classes should have the same row number designations in case of all models/matrices. E.g. Fig. 8 (h), (i), (j) row 0 include 35 hits while other confusion matrices in Fig 8 include 35 hits in row number 3. Such changes in class identifiers make this figure hard to understand.

This problem should be removed.

Authors gave the explanation within their response to the Reviewer.

The explanation should be added within the text of the manuscript it must be available to the Reader.

Response: We added these lines in the manuscript according to the reviewer suggestion.

Figures 7 and 8 show the confusion matrix for both machine learning and deep learning models on the same size of test data. We have done separate data splitting (Training and Testing sets) for machine learning and deep learning models. When splitting the dataset, we used the shuffle parameter which changes the number of instances for each target class for both machine and deep learning models but in both cases, the total number of test samples is the same which is 230. We did splitting separately for deep learning models because you can see the last layer of deep learning has 5 neurons so we have to feed target data after converting it into five variables we did splitting again after the conversion of data for deep learning models and shuffle parameters change the count of for each category in the test dataset.

Concern 10: 3.3 Sums of the same rows (the same classes) are not equal between matrices in Fig 7. If this is not an error then it should be explained in more clear way.

Authors gave the explanation within their response to the Reviewer.

The explanation should be added within the text of the manuscript it must be available to the Reader.

Response: This is the same concern as above and we explain it in the text. This difference in the number of instances in each row is because of reshuffle parameters in the train test split. This shuffle makes validation more strong when you have an imbalance or a small dataset.

We added an explanation for this in the manuscript as above for readers.

---

## [Decision Letter · Decision Letter 4]

27 Sep 2023

PONE-D-22-32955R4SSC: The Novel Self-Stack Ensemble Model for Thyroid Disease PredictionPLOS ONE

Dear Dr. Ji,

Thank you for submitting your manuscript to PLOS ONE. It was analyzed by me as an Academic Editor (Reviewer #4). After careful consideration, we feel that it has merit but does not fully meet PLOS ONE’s publication criteria as it currently stands. Therefore, we invite you to submit a revised version of the manuscript that addresses the points raised during the review process. In particular:reporting of statistical results should be improved,selection of statistical tests should be backed up by the presented data,indicated presentation and language problems should be removed.Please submit your revised manuscript by Nov 11 2023 11:59PM. If you will need more time than this to complete your revisions, please reply to this message or contact the journal office at plosone@plos.org. Please include the following items when submitting your revised manuscript:A rebuttal letter that responds to each point raised by the academic editor and reviewer(s). You should upload this letter as a separate file labeled 'Response to Reviewers'.A marked-up copy of your manuscript that highlights changes made to the original version. You should upload this as a separate file labeled 'Revised Manuscript with Track Changes'.An unmarked version of your revised paper without tracked changes. You should upload this as a separate file labeled 'Manuscript'.If applicable, we recommend that you deposit your laboratory protocols in protocols.io to enhance the reproducibility of your results. Protocols.io assigns your protocol its own identifier (DOI) so that it can be cited independently in the future. For instructions see: https://journals.plos.org/plosone/s/submission-guidelines#loc-laboratory-protocols. Additionally, PLOS ONE offers an option for publishing peer-reviewed Lab Protocol articles, which describe protocols hosted on protocols.io. Read more information on sharing protocols at https://plos.org/protocols?utm_medium=editorial-email&utm_source=authorletters&utm_campaign=protocols.

We look forward to receiving your revised manuscript.

Kind regards,

Maciej Huk, Ph.D.

Academic Editor

PLOS ONE

Journal Requirements:

Reviewers' comments:

Reviewer's Responses to Questions

**Comments to the Author**

1. If the authors have adequately addressed your comments raised in a previous round of review and you feel that this manuscript is now acceptable for publication, you may indicate that here to bypass the “Comments to the Author” section, enter your conflict of interest statement in the “Confidential to Editor” section, and submit your "Accept" recommendation.

Reviewer #4: (No Response)

2. Is the manuscript technically sound, and do the data support the conclusions?

Reviewer #4: Partly

3. Has the statistical analysis been performed appropriately and rigorously? 

Reviewer #4: I Don't Know

4. Have the authors made all data underlying the findings in their manuscript fully available?

Reviewer #4: Yes

5. Is the manuscript presented in an intelligible fashion and written in standard English?

Reviewer #4: Yes

6. Review Comments to the Author

Reviewer #4: >>> 1. Language problems:

1.1 Authors write: "We did splitting separately for deep learning models because you can see the last

layer of deep learning has 5 neurons so we have to feed target data after converting it

into five variables we did splitting again after the conversion of data for deep learning

models and shuffle parameters change the count of for each category in the test dataset."

Authors should not mix plain language with scientific language ("you can see").

Additionally - this sentence should be split into shorter and more clear ones. The current form is hard to understand.

>>> 2. Presentation problems:

Fig 10. (a) vertical axis title is improper (names of classifiers are under horizontal axis)

Fig 10. (b) vertical axis title is not as title of the figure suggests.

>>> 3. Other problems:

3.1 Reporting of statistical analysis is not complete:

- what is the sample size? (How many times each measurement was repeated? How many times 10-fold CV was repeated?)

- Authors report statistical importance, but what is the size of the observed effect (e.g. Cohens delta?) ?

- What is the estimated power of the performed test? (knowing the Cohens delta and number of samples you can use e.g. "Statistical Nomograph for Sample Size Estimation by Richard N. MacLennan"). Power of the test should be not lower than 0.7. This will allow to check if the number of measurments (sample size) is big enough.

3.2 Selection of the statistical test is not backed up by the presented data and Authors do not present information if the analysed data meet assumptions needed to use this test (normal distribution, homogeneity of variance). What statistical tests of normality and variance homogeneity were used to validate this?

3.3 Construction of the statistical test seems to be not valid. Authors are using solution which is typically used as a post-hoc test. Maybe Authors could use Friedmans ANOVA as a main test?

Please see e.g. [A] and [B] for details.

[A] Janez Demsar, Statistical Comparisons of Classifiers over Multiple Data Sets, Journal of Machine Learning Research 7, 2006

[B] Dietterich T.G., Approximate statistical tests for comparing supervised classification learning algorithms, Neural Comput 10(7), 1998

>>> Recommendation: minor rework.

7. PLOS authors have the option to publish the peer review history of their article (what does this mean?). If published, this will include your full peer review and any attached files.

Reviewer #4: No

---

## [Author Response · Author response to Decision Letter 4]

11 Oct 2023

Editor Comments:

Thank you for submitting your manuscript to PLOS ONE. It was analyzed by me as an Academic Editor (Reviewer #4). After careful consideration, we feel that it has merit but does not fully meet PLOS ONE’s publication criteria as it currently stands. Therefore, we invite you to submit a revised version of the manuscript that addresses the points raised during the review process.

In particular:

• reporting of statistical results should be improved,

We have updated the manuscript by incorporating the reviewer's suggested details on the power of the test.

• selection of statistical tests should be backed up by the presented data,

The data used for the test is already in the manuscript, as we utilized the model evaluation scores in the statistical analysis. So every model evaluation is present in the results section.

• indicated presentation and language problems should be removed.

We thoroughly reviewed the manuscript and corrected issues according to reviewer suggestions.

Reviewer #4:

>>> 1. Language problems:

1.1 Authors write: "We did splitting separately for deep learning models because you can see the last

layer of deep learning has 5 neurons so we have to feed target data after converting it

into five variables we did splitting again after the conversion of data for deep learning

models and shuffle parameters change the count of for each category in the test dataset."

Authors should not mix plain language with scientific language ("you can see").

Additionally - this sentence should be split into shorter and more clear ones. The current form is hard to understand.

Response: We would like to thank the reviewer for their valuable comments and suggestions. The manuscript has been updated in accordance with these suggestions, and the changes can be found in the updated version

>>> 2. Presentation problems:

Fig 10. (a) vertical axis title is improper (names of classifiers are under horizontal axis)

Fig 10. (b) vertical axis title is not as title of the figure suggests.

Response: The reviewer raised valuable points in the manuscript, and we have updated the manuscript to address their concerns. Updated figures can be found in the revised manuscript.

>>> 3. Other problems:

3.1 Reporting of statistical analysis is not complete:

- what is the sample size? (How many times each measurement was repeated? How many times 10-fold CV was repeated?)

- Authors report statistical importance, but what is the size of the observed effect (e.g. Cohens delta?) ?

- What is the estimated power of the performed test? (knowing the Cohens delta and number of samples you can use e.g. "Statistical Nomograph for Sample Size Estimation by Richard N. MacLennan"). Power of the test should be not lower than 0.7. This will allow to check if the number of measurments (sample size) is big enough.

Response: We would like to express our gratitude to the reviewer for their valuable comments. To determine the statistical power of our test, we utilized the suggested methods and an online power calculator [1]. Our sample size is 16, Cohen's effect size is 0.8, and the significance level is set at 0.05. The sample population consists of a normal distribution. The test results provide a statistical power of 0.169, with critical values of -1.96 and 1.96 based on the standard normal distribution, where the standard distribution shift is 2.263.

1- https://www.statskingdom.com/32test_power_t_z.html

3.2 Selection of the statistical test is not backed up by the presented data and Authors do not present information if the analysed data meet assumptions needed to use this test (normal distribution, homogeneity of variance). What statistical tests of normality and variance homogeneity were used to validate this?

Response: We used a normal distribution for the test, and all other parameters in the test were included in the updated manuscript. However, the reviewer raised concerns regarding the quality of the data, as we utilized model results for performing a statistical T-test. This means that the results from the models, including accuracy, per-class (precision, recall, F1 score), and their average, were subjected to a T-test to demonstrate their significance. We compared the results of the SSC model with those of all other models, as displayed in Table 15.

3.3 Construction of the statistical test seems to be not valid. Authors are using solution which is typically used as a post-hoc test. Maybe Authors could use Friedmans ANOVA as a main test?

Please see e.g. [A] and [B] for details.

[A] Janez Demsar, Statistical Comparisons of Classifiers over Multiple Data Sets, Journal of Machine Learning Research 7, 2006

[B] Dietterich T.G., Approximate statistical tests for comparing supervised classification learning algorithms, Neural Comput 10(7), 1998

Response: We would like to express our gratitude to the reviewer for their valuable comments. Indeed, the reviewer's point is accurate, as we deployed a T-test on the model results to assess the statistical differences in model performance. The study [B] recommended applying these tests to model predictions, which can help obtain a larger sample for statistical analysis. We conducted statistical tests on the evaluation scores to emphasize the significance of the models over others based on their performance scores. In this study, our main contribution is classification and proposing a model. Our aim is to highlight the significance of the proposed models and we adopt a very simple approach to apply statistical Tests rather than adopting complex approach.

---

## [Decision Letter · Decision Letter 5]

23 Nov 2023

SSC: The Novel Self-Stack Ensemble Model for Thyroid Disease Prediction

PONE-D-22-32955R5

Dear Dr. Ji,

We’re pleased to inform you that your manuscript has been judged scientifically suitable for publication and will be formally accepted for publication once it meets all outstanding technical requirements.

Kind regards,

Maciej Huk, Ph.D.

Academic Editor

PLOS ONE

Additional Editor Comments (optional):

Reviewers' comments:

Reviewer's Responses to Questions

**Comments to the Author**

1. If the authors have adequately addressed your comments raised in a previous round of review and you feel that this manuscript is now acceptable for publication, you may indicate that here to bypass the “Comments to the Author” section, enter your conflict of interest statement in the “Confidential to Editor” section, and submit your "Accept" recommendation.

Reviewer #4: All comments have been addressed

2. Is the manuscript technically sound, and do the data support the conclusions?

Reviewer #4: Yes

3. Has the statistical analysis been performed appropriately and rigorously? 

Reviewer #4: I Don't Know

4. Have the authors made all data underlying the findings in their manuscript fully available?

Reviewer #4: Yes

5. Is the manuscript presented in an intelligible fashion and written in standard English?

Reviewer #4: Yes

6. Review Comments to the Author

Reviewer #4: >>> 1. Language problems: not detected

>>> 2. Presentation problems:

2.1 Fig. 9, Fig. 10 a/b - please add units to description of vertical axis. (e.g. accuracy equal 0.9 can mean 90% or 0.9%). Do Authors mean "Accuracy [1]" ?

2.2 Fig. 6(f) - description of vertical axis is not visible

>>> 3. Other problems:

3.1 ref. [2] (https://weillcornell.org/news/understanding-thyroid-problems-disease) Such source is not verified and not stable - not a proper reference within scientific manuscript. Authors should use reference to related, high quality book or scientific article.

3.2 ref. [52] - please see comment 3.1 above.

>>> Recommendation: Accept after fixing abovementioned minor problems

7. PLOS authors have the option to publish the peer review history of their article (what does this mean?). If published, this will include your full peer review and any attached files.

Reviewer #4: No

---

## [Editor Report · Acceptance letter]

20 Dec 2023

PONE-D-22-32955R5 

PLOS ONE

Dear Dr. Ji, 

I'm pleased to inform you that your manuscript has been deemed suitable for publication in PLOS ONE. Congratulations! Your manuscript is now being handed over to our production team.

Kind regards, 

on behalf of

Dr. Maciej Huk 

Academic Editor

PLOS ONE